# High-speed imaging of light-induced photoreceptor microsaccades in compound eyes

Joni Kemppainen [1], Neveen Mansour[1], Jouni Takalo[1] & Mikko Juusola [1,2] ✉

Inside compound eyes, photoreceptors contract to light changes, sharpening retinal images of the moving world in time. Current methods to measure these so-called photoreceptor microsaccades in living insects are spatially limited and technically challenging. Here, we present goniometric high-speed deep pseudopupil (GHS-DPP) microscopy to assess how the rhabdomeric insect photoreceptors and their microsaccades are organised across the compound eyes. This method enables non-invasive rhabdomere orientation mapping, whilst their microsaccades can be locally light-activated, revealing the eyes' underlying active sampling motifs. By comparing the microsaccades in wild-type *Drosophila*'s open rhabdom eyes to *spam*-mutant eyes, reverted to an ancestral fused rhabdom state, and honeybee's fused rhabdom eyes, we show how different eye types sample light information. These results show different ways compound eyes initiate the conversion of spatial light patterns in the environment into temporal neural signals and highlight how this active sampling can evolve with insects' visual needs.

[1] Department of Biomedical Science, University of Sheffield, Sheffield S10 2TN, UK. [2] National Key Laboratory of Cognitive Neuroscience and Learning, Beijing, Beijing Normal University, 100875 Beijing, China. ✉email: m.juusola@sheffield.ac.uk

Because the insect compound eyes extend from the rigid head exoskeleton, appearing stationary to an outside observer, it was long assumed that their inner workings would also be static. Therefore, as the eyes' ommatidial faceting sets their photoreceptor spacing, it was deduced that the compound eyes could only sample a pixelated low-resolution image of the world[1–3].

However, recent results on the *Drosophila* compound eyes are now replacing this *static* viewpoint with a new concept of morphodynamic *active* sampling[4,5]. Sophisticated experiments have revealed how photoreceptor microsaccades *locally*[4,5] (Fig. 1a) and intraocular muscle contractions *globally*[4–7] move, stretch and recoil intraommatidial optical structures, improving vision morphodynamically. During the *local ultrafast* (<100 ms) *photomechanical* microsaccades, the photoreceptors of a single ommatidium concurrently recoil axially (Fig. 1a, left) and swing laterally (right) to increase sampling resolution in space and sharpen light input in time for super-resolution vision[4]. And, with the left and right eye photoreceptor pairs generating mirror-symmetric microsaccades, this active sampling further expands the flies' hyperacute stereopsis[5]. Conversely, the intraocular

muscle contractions shift one eye's entire retina (its sampling matrix) *globally* regarding the other eye[4–7]. In head-immobilised *Drosophila*, these drifts and vergence movements, which also happen underneath the eyes' rigid ommatidial lens cover, hidden from the outside view, are typically 10 to 100 times slower than the local photoreceptor microsaccades[4,5]. But in freely behaving flies[6], their dynamics may strengthen to combat adaptive perceptual fading[4,7] and contribute to attentive saccadic viewing and object tracking[6].

Minute photomechanical photoreceptor contractions (<~200 nm) were first measured in ex vivo *Drosophila* preparations using atomic force microscopy (AMF)[8]. Initially, these movements, caused by PIP$_2$ cleavage in the microvillar photoreceptor membrane[8,9], were thought to be too small to alter the photoreceptors' light input[8]. However, later live-microscopy experiments[4], using the cornea neutralisation method[10], showed that ex vivo AFM underestimates the size of the lateral rhabdomere movements[4]. Intense light modulation in vivo could rapidly swing an R1-R6 rhabdomere about its width (~1400 nm) sideways. And with similar microsaccades also occurring in synaptically decoupled photoreceptors[4], the results demonstrated active

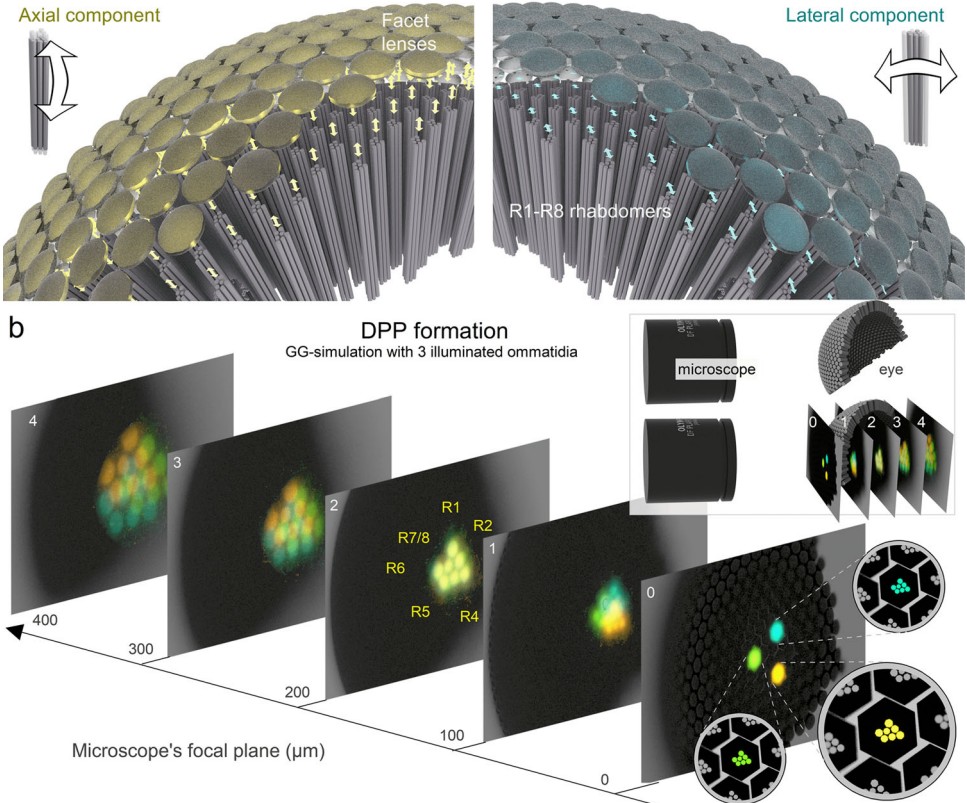

**Fig. 1 Active sampling by photomechanical photoreceptor microsaccades and the deep pseudopupil phenomena. a** In *the conventional static sampling theory*[1,2], ommatidial facets set a compound eye's photoreceptor spacing, limiting the finest image details the eye could resolve. However, inside an ommatidium, incoming light intensity changes make its R1-R7/8 photoreceptors rapidly recoil axially and swing laterally. These so-called ultrafast photoreceptor microsaccades enable *Drosophila* to see the world in a finer resolution than its eyes' photoreceptor spacing, explained by *the new active sampling theory*[4,5]. Left: *Drosophila* eye computer graphics (CG) model highlights the axial microsaccade component; R1-R7/8s first recoil and then slide towards the ommatidium lens. Right: concurrently, the light-activated R1-R7/8 also swing sideways (laterally). A local (incident) light intensity change evokes microsaccades only in those ommatidia facing the stimulus. If this happens in the frontal left and right eye ommatidia with overlapping receptive fields, their microsaccades are synchronous yet have mirror-symmetric lateral components[4,5]. Meanwhile, elsewhere across the eyes, the photoreceptors stay still because the eye curvature and the ommatidial screening pigments block them from seeing the stimulus[4,5]. **b** The optical principle of the deep pseudopupil (DPP). DPP is a virtual image of several distal R1-R7/8 rhabdomere tips (highlighted in blue, yellow and green for three nearby ommatidia), which align with the angle the eye is observed at while being ~10×-magnified by the ommatidial lens system. These virtual rhabdomere images are optically brought together when the microscopes' focal plane is ~200 μm under the eye surface (as shown in image 2). Because of the optical magnification, the rhabdomere tips, which appear deep inside the eye, are actually positioned at ~20 μm from the inner surface of the ommatidium lens.

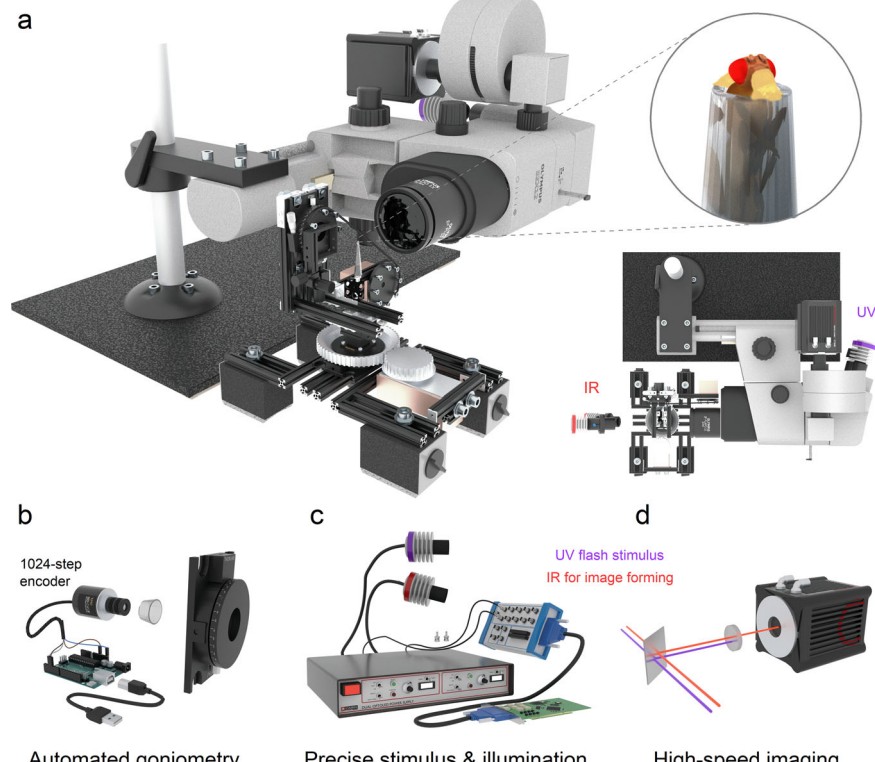

**Fig. 2 Goniometric high-speed deep pseudopupil (GHS-DPP) imaging system. a** Its integral parts are a sideways mounted stereomicroscope with a high-speed (100–1000 fps) digital camera and a goniometric rotation stage system. *Drosophila* eyes are imaged under the antidromic dual infrared illumination, invisible to the flies[4,25]. Photoreceptor microsaccades are activated by ultraviolet (UV or green) light stimulation, delivered through the ocular slot system. **b** The fly's *x/y*-rotations are read using a 1024-step rotary encoder and an Arduino board. **c** The high-power LEDs and the camera were controlled over the BNC interface. **d** An infrared (IR) passing but UV blocking optical filter in the front of the camera decouples the UV-stimulus from the imaging pathway. See Supplementary Video 1.

sampling inside an ommatidium for the first time[4]. Unfortunately, both these methods are technically demanding and spatially limiting and thus ill-suited for mapping the microsaccade movement sizes and directions across the left and right compound eyes.

In the *Drosophila* compound eyes, an optical phenomenon called the deep pseudopupil (DPP) arises from the regular arrangement of ommatidia (Fig. 1b), each containing R1-R7/8 photoreceptors in which open rhabdomeres act as waveguides[11]. By focussing a microscope's image plane below the eye's outer surface (Fig. 1b, inset), virtual images of several ommatidia's R1-R7/8 rhabdomere patterns (Fig. 1b, highlighted in blue, green and yellow for the nearby ommatidia) become superimposed, revealing their stereotypical yet ~10 times magnified trapezoid arrangement. And since these virtual images fuse at the microscope focal plane of ~200 $\mu$m, we see the rhabdomere tips inside the eye, ~20 μm away from the inner surface of their ommatidium lenses. Thus, DPP microscopy offers a versatile, non-invasive method to observe retinal tissue in living flies and other insects (Fig. 2). First, to observe a well defined, clear DPP pattern in dipteran eyes requires precisely organised rhabdomeres across the neighbouring ommatidia[12], and DPP microscopy with epi-illumination[13,14] (frontally, through the eye optics) has been used to study retinal degeneration that breaks this order[15–17]. Second, because the rhabdomeres contributing to the DPP image are those facing the observer, the DPP microscopy provides the "gold standard" measure for the binocular overlap over the left and right compound eyes[18–20]. Finally, because any lateral retinal tissue movement shifts the DPP similarly, DPP microscopy can be used to investigate how the eye-muscle-induced retinal micromovements shift the photoreceptors' receptive fields[5,21].

Here, we present a novel goniometric high-speed deep pseudopupil (GHS-DPP) microscopy (Fig. 2) with invisible (850 nm) infrared back-illumination, developed to study active sampling in insect compound eyes. We first use it to measure the photoreceptor microsaccade dynamics and directions in wild-type *Drosophila melanogaster*, possessing the archetypal open rhabdom dipteran eyes (Fig. 3, top), and transgenic *spam* null-mutants, in which rhabdomeres fail to separate (Fig. 3, middle), forming an ancestral, fused rhabdom (apposition) eye[22]. Finally, we measure photoreceptor microsaccades in the Honeybee (*Apis mellifera*) apposition eyes (Fig. 3, bottom) and compare these dynamics to wild-type and *spam Drosophila*. Our results show that active sampling by photoreceptor microsaccades occurs both in the open and fused rhabdom eyes. We analyse their functional similarities and differences and discuss what these results could mean for the evolution of compound eyes and insect vision in general[23].

## Results

**Left and right eye DPP patterning rotate systematically and mirror-symmetrically.** We first inspected the wild-type and *spam* flies' DPP patterns in still images taken under antidromic infrared illumination in perceptual darkness (Fig. 4). Characteristically, the wild-type DPP image merged the neighbouring ommatidia's R1-R7/8 rhabdomere images into a classic trapezoid pattern of seven small bright discs (Fig. 4a). In contrast, the *spam* DPP appeared as a tiny bright disc (Fig. 4d). These different DPP patterns were entirely expected, as they directly follow the deep pseudopupil theory[12] (Fig. 1b) and are reproduced by our CG-modelling (Fig. 3).

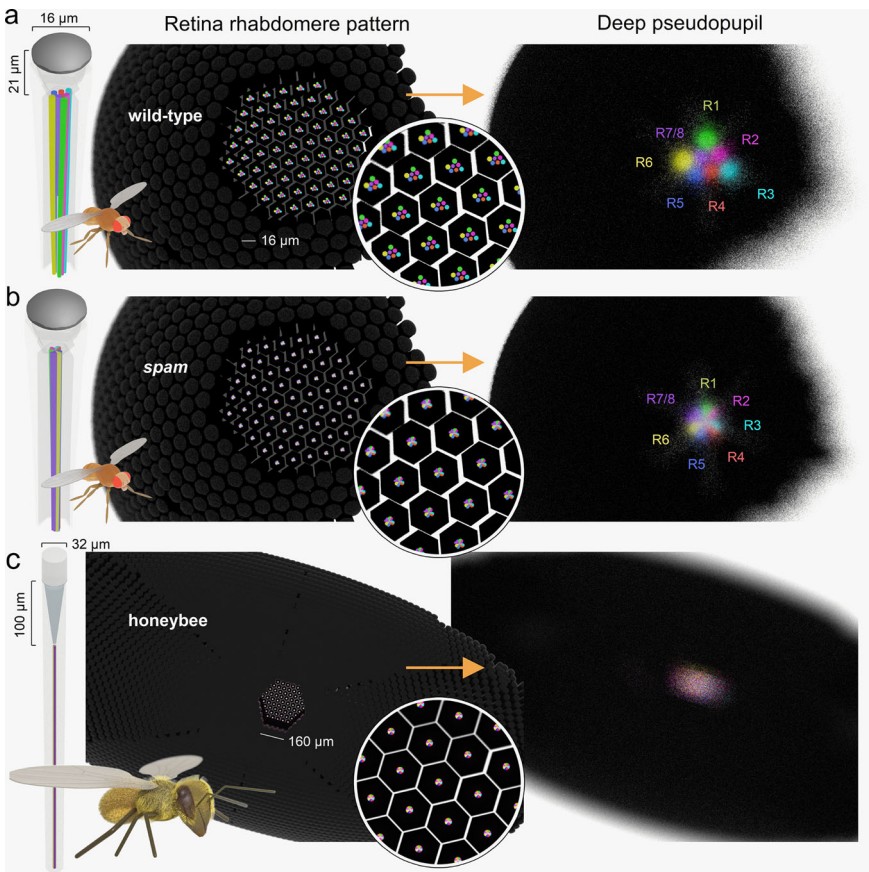

**Fig. 3 Deep pseudopupil (DPP) in the wild-type and *spam* *Drosophila* and honeybee apposition eyes. a** Left: the *Drosophila* compound eye comprises ~750 ommatidia, each having a light focusing facet lens and eight photoreceptors; with their light-sensitive parts, the rhabdomeres (coloured), protruding centrally. Middle: Computer graphics (CG) models of the open wild-type rhabdomeres; here shown for their left eye's southern hemisphere. Right: The CG-model simulated DPP image appears as a magnified "virtual" image of the open rhabdomere patterns. Here, DPP is shown at the focal plane of 200 µm into the eye for the right eye ventral section, corresponding to frame #2 in Fig. 1b and the bottom-left frame in Fig. 4b. **b** Left: in the *spam*, R1-R7/8 rhabdomeres reverted back into their ancestral fused rhabdom form. Middle: inside the ommatidia (in the retinal tissue), the fused rhabdom resembles a single rod. Right: their DPP appears as a tiny bright disc. Colouring indicates the contributing R1-R7/8 rhabdomeres' relative locations in the virtual DPP rhabdom image. **c**, Left and Middle: In the honeybee apposition eye ommatidia, the photoreceptor rhabdomeres form a single rod-like waveguide, the rhabdom, centred ~100 µm behind the ommatidial lens. Right: In the DPP image, the honeybee fused rhabdom appears as a single disk, similar to *spam*.

Because *Drosophila* has a small head and fairly transparent (not densely pigmented) cuticle, the infrared GHS-DPP microscopy makes it straightforward to record, measure and map the DPP pattern changes across the eyes. Specifically, for *Drosophila*, we could use relatively low-power infrared illumination—propagating through its intact head—and still obtain high signal-to-noise DPP imaging at high frame rates (≥100 fps). The real benefit was that since the tested flies required no surgical cuticle removal (to improve infrared throughput for better DPP image quality) and suffered very little or no heat damage, individual *Drosophila* regularly provided consistent, repeatable results throughout hours-long experiments.

In the wild-type, the left and right eyes' DPPs are mirror-symmetric: shown in the binocular upper-frontal view (Fig. 4a; the blue and red boxes highlight their right and left DPPs, respectively). Interestingly, the north and south hemispheres of the eye also have mirror-symmetric DPP patterns[24] but fuse at the equator (midline) to form larger elongated triangle shapes (Fig. 4b). The DPP orientation, as the angular rotation between R3–R2–R1 rhabdomeres (yellow line) and R3–R4–R5 rhabdomeres (green), shifts between nearby eye locations in regular small steps, generating the left and right eyes' mirror-symmetric global map (Fig. 4c). In this global map, local DPP alignments follow a concentrically expanding diamond-shaped pattern.

Therefore, the underlying developmentally rotated R1-R7/8 rhabdomere orientations at each eye position align with the frontally expanding optic flow field[5] (Supplementary Video 3). In contrast, in the *spam* mutant, because their DPPs appear as homogeneous circular or oval discs (Fig. 4d), neither such gradual rotations nor their left-right and north-south divisions were readily observable (Fig. 4e).

**Photoreceptor microsaccades' lateral and axial components**. After imaging the eyes' DPPs, we tested whether the *spam* eyes can generate ultrafast (time-to-peak < 100 ms) photomechanical microsaccades, akin to the open rhabdom wild-type[4,5] eyes. In the first instance (Fig. 5a, b), these experiments were performed at the fixed left and right eye locations (±28° horizontal, −37° vertical) using a bright 200-ms-long ultraviolet (365 nm) flash stimulus (Supplementary Video 2). In each fly, the UV flash was delivered locally at the centre of the observed DPP photoreceptors' receptive fields (Supplementary Video 4) and repeated 25 times to obtain robust estimates and statistics of the resulting response dynamics.

A flash of any wavelength R1-R8 photoreceptors are sensitive to (~300 to ~650 nm)[25,26] evokes a photoreceptor microsaccade[5]. Inside an ommatidium, the number of light-activated photoreceptors and their combined contraction strength set its

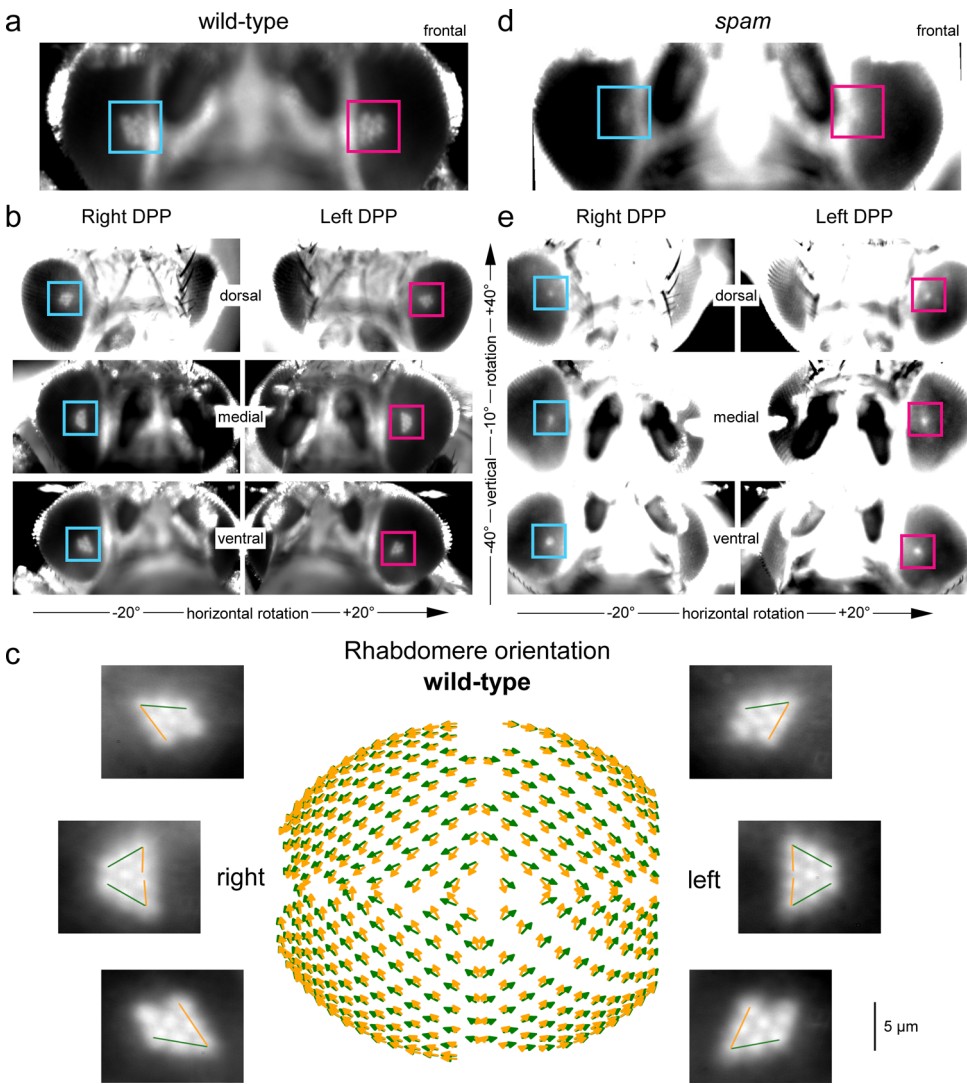

**Fig. 4 Wild-type and *spam* DPP imaged by the 850 nm GHS-DPP microscopy. a** Forward-facing wild-type eyes (horizontal rotation = 0°) with the red and blue square boxes indicating their left and right DPPs, respectively. **b** Wild-type eyes, viewed on the dorsal side (1st row), at the midline (2nd row), and on the ventral side (3rd row), reveal their DPPS ventral-dorsal symmetry. **c** The wild-type rhabdomere orientation across the left and right eyes as mapped with the GHS-DPP microscopy. The green lines indicate the R1-R2-R3 rhabdomere axes, and the yellow lines the R3-R4-R5 axes. R1-R7/8 rhabdomeres rotate systematically with changing eye location, producing concentrically expanding diamonds-patterns[5], highlighting the characteristic right-left eye and north-south (dorsal-ventral) hemisphere mirror-symmetricity. **d** The left and right eye *spam* DPPs appear circular-symmetric discs and are more oversaturated due to their eyes' lighter pigmentation. **e** *Spam* eyes with their DPPs are viewed on the dorsal side, midline, and ventral side.

photoreceptor microsaccade amplitude[5]. Because these photoreceptors are mechanically coupled and likely pivoted[5], it only takes one (say R1) to be light-activated, and its contraction alone can move its neighbours (R2–R8) too[5] (Supplementary Video 4). As *Drosophila* R1-R6 possesses the sensitising UV-pigment and R7s are UV-sensitive[25,26], a UV flash evokes larger photoreceptor microsaccades than, say, an amber-flash, which only activates R8y cells[5]. Therefore, in *Drosophila* experiments, UV flash is a good choice of stimulus.

In wild-type, a UV flash always evoked a local photomechanical photoreceptor microsaccade, making the observed DPP rapidly jump laterally in the front-to-back (north-west) direction (Fig. 5c) before swiftly returning in darkness, as expected for normal eye function[4,5]. However, remarkably, we found that the fused rhabdom *spam* eyes also generate robust ultrafast DPP microsaccades (Fig. 5d) with broadly similar temporal dynamics. In both phenotypes, besides moving laterally (Fig. 5d, above), the photoreceptors moved simultaneously also axially (below). This

axial component, resulting from the rhabdomeres contracting away and elongating towards the ommatidium lens (and the camera)[5], can be directly measured as a proportional DPP darkening and lightening[5]. To eliminate any motion artefacts, we measured the axial component from the DPP image pixels' dynamic intensity change, tracking frame by frame only the pixels within the rhabdomere tips. As expected, the fast axial DPP brightness changes systematically time-locked with the corresponding lateral DPP movements (Fig. 5d–f), consistent with both phenotypes having the same photomechanical phototransduction origin.

Qualitatively, both the wild-type and *spam* had similar looking microsaccade kinematics and probabilities (Fig. 5d–f), but the overall displacement amplitudes appeared much smaller in *spam* (Fig. 5d). Maximum amplitude measurements confirmed that the *spam* DPP microsaccades (both their lateral and axial components) were indeed smaller than the wild-type (Fig. 5g). Similarly, the *spam* flies' calculated maximum microsaccade activation

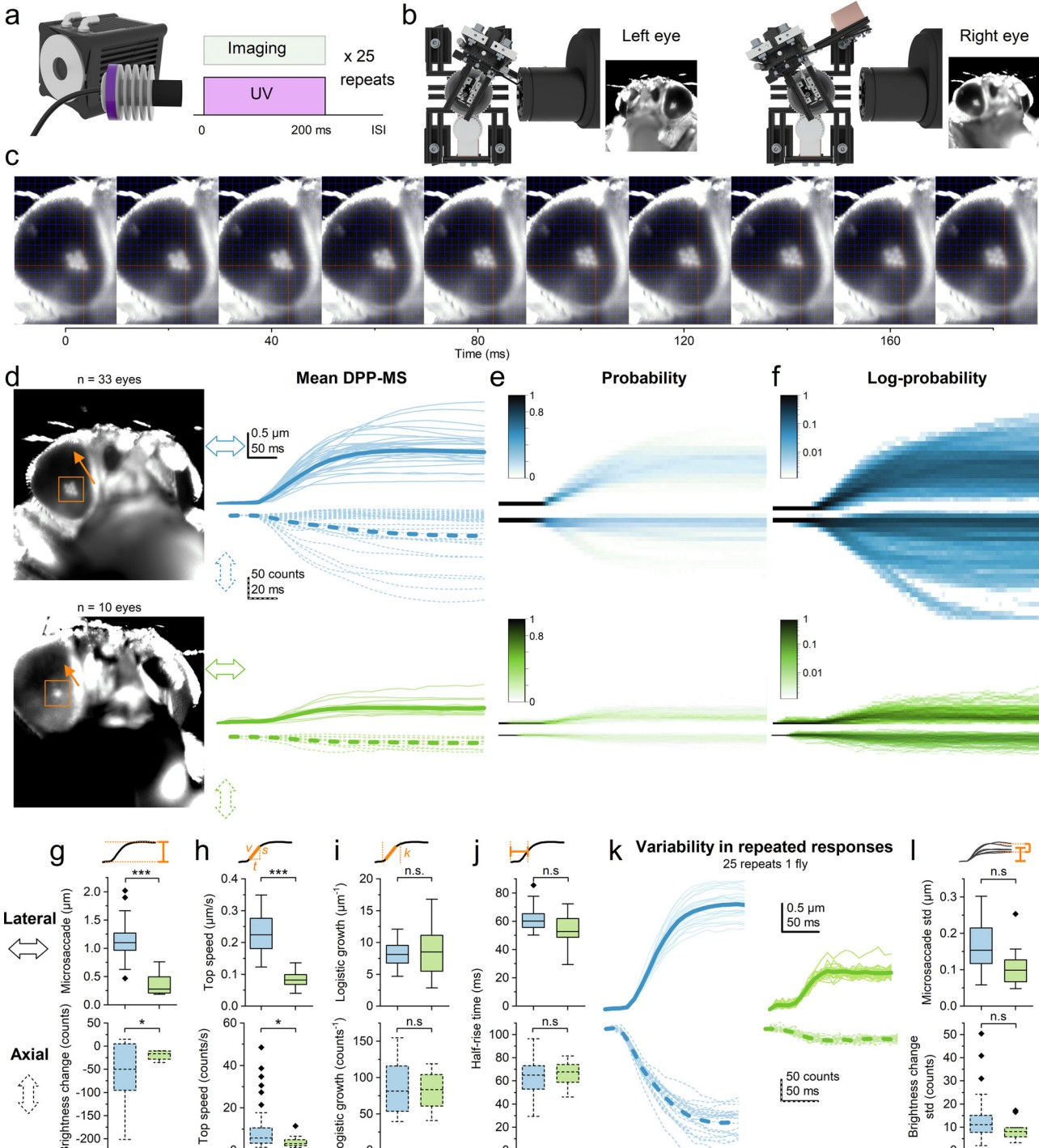

**Fig. 5 DPP microsaccade kinematics in the wild-type and *spam*. a** In the local recordings, the UV flash was repeated 25 times following a regular inter-stimulus interval (ISI). **b** These experiments were done at one fixed location in each eye. **c** Image time series from the stimulus onset (time zero) to 180 ms later, showing how the same DPP pattern, plotted against fixed red *x–y*-coordinate axes, moves back-to-front (North-West) about one rhabdomere width (~1.5 mm inside ommatidia). **d** Wild-type and spam DDP patents and their tracked movement directions (orange arrows). Mean rhabdomere lateral (continuous traces) and axial (fast DPP darkening/brightening, dotted) displacement components in the wild-type (1st row) and *spam* (2nd row) appear similar in shape but are smaller in *spam*. The thick lines show the corresponding population means. **e**, **f** The probability plots, containing the separate (single) responses, indicate that the photoreceptor microsaccades occur over a predictable but variable range. **g–j** The microsaccade amplitude, the top speed, the logistic growth factor and the half-rise time, respectively, in the wild-type and *spam* (Welch's *t* test; $p = 0.027$, $p = 0.031$, $p = 0.86$, $p = 0.62$). **k** The DPP microsaccade responses of one selected wild-type and one *spam* show variability in repeated responses that may indicate purposeful modulation. **l**, Quantified displacement standard deviations (std) for all the wild-type and *spam* flies do not differ statistically (Welch's *t* test; $p = 0.16$).

speed was significantly slower (Fig. 5h). However, these phenotypes' logarithmic growth factors during the activation phase did not differ statistically (Fig. 5i), meaning that the *spam* DPP microsaccades' slower speed resulted from their shorter travelling distance but not from changes in their duration. Accordingly, there was no significant change in the half-rise times (Fig. 5j). The similar logarithmic growth factors and half-rise times suggest that the likely photomechanical cause of the DPP microsaccades, the $PIP_2$ cleavage from the microvillar membrane during phototransduction[8], is unaffected by the *spam* mutation. Overall, these results show that the *spam* flies have similar microsaccade kinematics as the wild-type but are only smaller.

We further inspected individual flies' DPP microsaccade variations to the 25 times repeated light flashes. A qualitative comparison between the selected wild-type and *spam* flies showed that most variation occurs in the total microsaccade size; that is, how far the DPP and the rhabdomeres travel during the activation phase (Fig. 5k). Furthermore, we found no difference between their DPP displacements' standard deviations (Fig. 5l), indicating similar intrinsic amplitude jitter, or photomechanical stochasticity, affecting wild-type and *spam* microsaccades. These results highlight that DPP microsaccades vary considerably between the flies and between individual flies' repeated responses, suggesting that their kinematics may further reflect top-down regulation from the brain[27,28], i.e. the flies internal state (circadian rhythm, attention or activity state)[27,29,30]. Such variability could result, for instance, from the slow eye-muscle-induced whole retina drifting[5], structurally fluctuating the local rhabdomere tension (their anchoring and pivoting dynamics) from one trial to another[5]. Notably, however, the real benefit of the local photoreceptor microsaccades' stochastic variability, irrespective of its cause, is that it effectively removes aliasing from the retinal sampling matrix[4,5,31].

**Mapping photoreceptor microsaccades' movement directions across the eyes**. Next, we mapped DPP microsaccades and their directions from about 200 distinct locations over the left and right eyes (Fig. 6a, b; Supplementary Video 5). In the wild-type, the DPP microsaccades moved approximately from back-to-front or south-to-north, depending on the eye region (Fig. 6c). But we did not image the DPP microsaccade relaxation phase, which occurs during light decrements (darkening), moving slower and in opposite directions to the activation phase[4,5] (returning rhabdomeres to their initial starting positions).

Interestingly, we discovered that the fused rhabdom *spam* DPP microsaccades generally moved along similar directions to those of wild-type (Fig. 6d). We calculated the absolute angular difference in the rotation of the microsaccades between the wild-type and *spam* flies and found their resulting global maps broadly similar (Fig. 6e; Supplementary Video 5). However, in a more detailed inspection, the *spam* left eye DPP microsaccades on the anterior parts showed slightly more counterclockwise rotation and the right eye DPP microsaccades slightly more clockwise rotation than the wild-type (Fig. 6f).

These results demonstrate active sampling—by photomechanical photoreceptor microsaccades—occurring in a spatially-coordinated manner across the *spam* fused rhabdom eyes. Moreover, since the wild-type microsaccade directions (their forward-and-back movement axes) align with their R1-R2-R3 rhabdomere axis (Fig. 3c, yellow arrows)[5], it seems highly likely that the *spam* R1–R8 photoreceptors inside the ommatidia would also rotate in an eye-position-dependent manner, resulting in the ventral-dorsal hemispheric DPP mirroring. However, we could not directly confirm this from the round and oval *spam* DPP images (Fig. 3e). Nevertheless, in both phenotypes, the

photoreceptors' *photomechanical* active sampling makes their receptive fields scan the world in their eye-location-specific directions, broadly matching the flies' concentrically expanding optic flow field during the forward locomotion[5] (Supplementary Video 5).

**Photoreceptor microsaccades in honeybee fused rhabdom apposition eyes**. Our current results, together with those from the previous studies[4,5,8,9], indicate that photomechanical transduction reactions within *Drosophila* photoreceptors are responsible for their ultrafast light-induced microsaccades. As these $PIP_2$-bound reaction steps[8] are thought to be evolutionarily conserved in rhabdomeric photoreceptors[32,33], we next tested whether the Honeybee (Fig. 7) fused rhabdom apposition eyes also generate photomechanical photoreceptor microsaccades.

The honeybee eyes and head are about 10 times larger than *Drosophila*'s and more densely pigmented, thicker, and sturdier. With long hairs, mouthparts and antennae sticking out, the bee head looks striking, like a 16th-century knight's helmet (Fig. 7a). Unfortunately, these structural features complicate GHS-DPP microscopy. The head's thick pigmented cuticle filters out infrared light and the antidromic light path (from the back of the head to the rhabdomere tips) is ~10 times longer than in *Drosophila*. Therefore, we had to surgically remove parts of the rear head cuticle and use a more powerful and condensed infrared beam to achieve sufficiently-high signal-to-noise ratio DPP images of the back-illuminated rhabdoms (Fig. 7b). Once a honeybee was aligned correctly, a point-like DPP image of one superimposed rhabdom (Fig. 7b, frame #12), collected from the neighbouring ommatidia, emerged with the microscope focussing through the ommatidial surface into the eye. These DPP patterns matched our CG-model prediction (cf. Fig. 3) and appeared similar to those in the *spam* fused rhabdom eye (cf. Fig. 4e).

We learned through trial and error not to strive for the best spatial resolution in high-speed imaging experiments. Instead, we optimised the setup to enable high-speed in vivo imaging by balancing the infrared power and exposure time with the DPP image contrast and temporal resolution (Fig. 7c, left). This optimisation process was complex as the temporal resolution was essential to avoid blurring caused by the photoreceptor contractions. Too low infrared power or frame rate, and the small and fast photoreceptor microsaccades were undetectable from the noisy DPP images. Too high power, and the honeybee was near-instantly killed by the heat cooking its brain. However, with appropriate settings, we could repeatedly record bee photoreceptor microsaccades in vivo (Fig. 7c).

Honeybee fused rhabdom photoreceptors generated photomechanical microsaccades to both tested UV (Fig. 7c, middle) and green light flashes (right); the given examples were recorded after prolonged dark adaptation. Unsurprisingly, the DPP microsaccades were relatively small, with their maximum displacement range (≤1 μm) equating to ≤1° receptive field jumps in space, being close to our earlier prediction[5]. This prediction was based on the bee rhabdom's envisaged "rod-like" rigidity, ~1.9° receptive field half-width[34] and the ~1° interommatidial angle[1] (see "Discussion"). Notably, the green flash evoked on average 1.54 times larger microsaccades than the UV flash. The greater green-sensitivity is consistent with each honeybee ommatidium (Fig. 7c, inset) having 4 large green-sensitive photoreceptors and 2.5 UV-sensitive ones (2 large photoreceptors at the opposite walls of each ommatidium + 1 small photoreceptor underneath at the base). In *Drosophila*, the number of light-activated photoreceptors and their combined contraction strength set the DPP microsaccade amplitude[5]. Thus, it seems

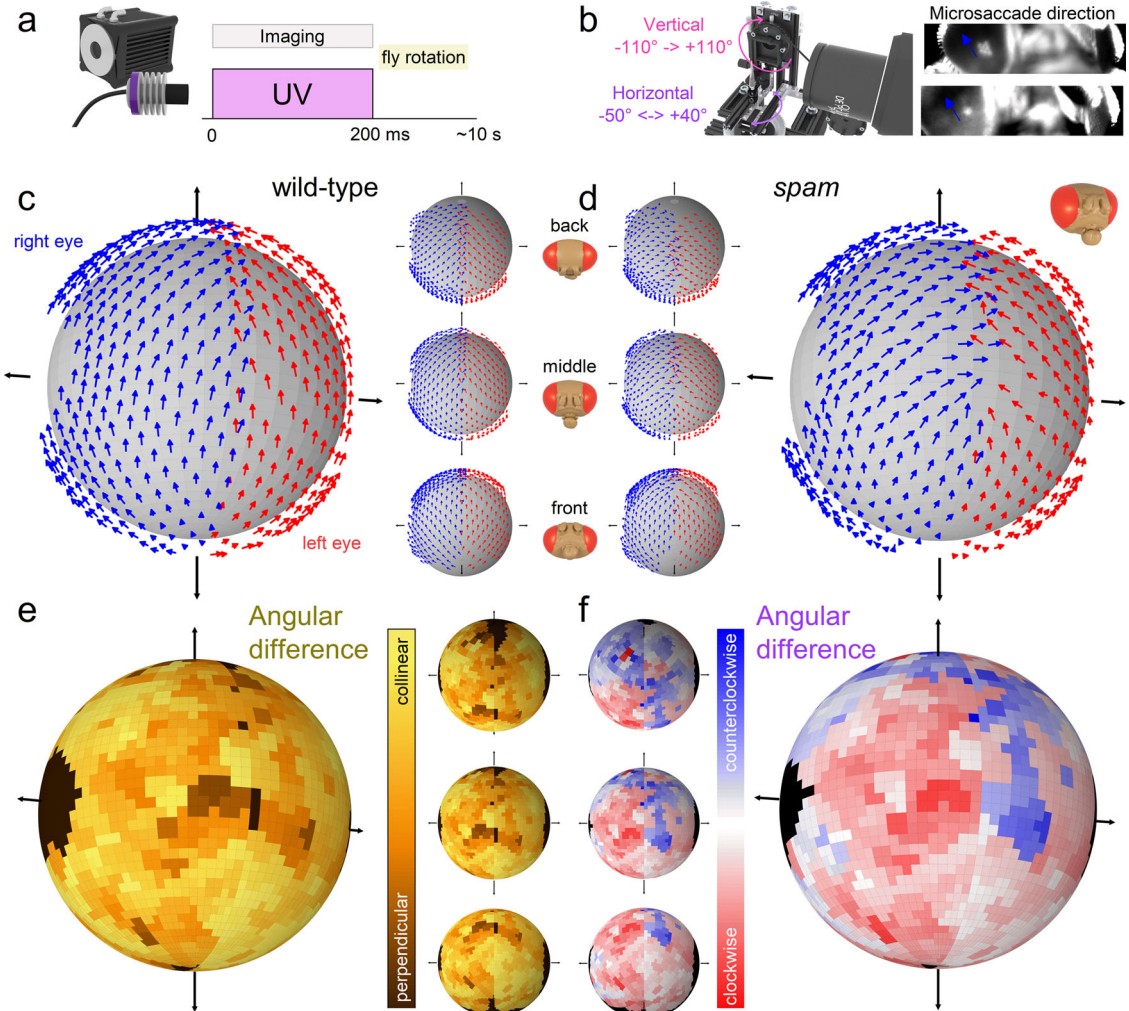

**Fig. 6 DPP microsaccades across the left and right wild-type and *spam* eyes. a** During the 200 ms UV flash, the camera recorded the DPP microsaccades 100 frames per second. **b** After each flash, the fly was rotated, recording approximately 200 locations on the left and right eyes. Blue arrows indicate DPP microsaccade directions. **c** Mean microsaccade vector map of wild-type flies ($N = 5$; the arrows indicate eye-location-specific normalised microsaccade directions). The small insets show the dorsal, the anterior and the ventral vector map views from the top to bottom. **d** Mean microsaccade vector map of spam files ($N = 5$) appears similar to the wild-type but only slightly noisier. **e** Angular difference plot of the wild-type and *spam* vector maps show that the wild-type and *spam* vector maps are mostly collinear. **f** On the anterior eye regions, the angular difference graph suggests that the *spam* right eye microsaccades are clockwise-rotated and the left eye microsaccades counterclockwise-rotated compared to the wild-type (as viewed from outside). See Supplementary Videos 2 and 5.

probable that similar additive photomechanics would govern the Honeybee DPP microsaccades too. Also expectedly, the microsaccades showed synchronised lateral and axial components, comparable to *Drosophila* (Fig. 5d-l).

Nevertheless, whilst indicative of their phototransduction origin, these recorded dynamics were somewhat slower than expected[5], with the photomechanical rhabdom movements reaching their maxima in about 80–150 ms (Fig. 7c), akin to *Drosophila* (Fig. 5). The probable explanation for this speed range is the prolonged dark adaptation, short 100 ms flashes (note, *Drosophila* was tested with 200 ms flashes, Fig. 5a) and 10 s inter-flash-intervals used in these experiments. After all, dark-adaption is well-known to decelerate honeybee phototransduction dramatically. Interestingly, however, the DPP microsaccades showed a consistent photomechanical transient (a nudge) ~30–50 ms from the light onset (black arrows). Such a nudge could, for example, signal a fast and large axial (inward) component, which DPP imaging with suboptimal resolution (of a relatively low signal-to-noise ratio) cannot resolve. Moreover, the microsaccade dynamics

varied greatly from trial to trial, even more so than in *Drosophila* (cf. Fig. 5k), suggesting that they could be modulated or influenced by intrinsic processes, such as intraocular muscles extruding a force on the ommatidial structures.

To test whether the photoreceptor microsaccade variability (Fig. 7c) could, in part, reflect spontaneous intraocular muscle activity drifting the entire retina (and thus potentially inflicting variable tension to the rhabdoms[5]), we next monitored honeybee DPP continuously in darkness (Fig. 7d). These long-term recordings lasted up to 16 minutes. The recordings showed slow wave-like lateral retina movements, occurring about 2–3 times in a minute, and gradual axial creep, almost certainly[5] pulling the observed rhabdoms inwards (DPP darkening). These spontaneous, presumably muscle-activity-induced, components (Fig. 7d) differed clearly from the ultrafast photomechanical photoreceptor microsaccades (Fig. 7c). They were largely unsynchronised in time, and most crucially, showed 10–100 times slower dynamics, broadly comparable to our earlier findings of intraocular muscle activity in *Drosophila*[4,5].

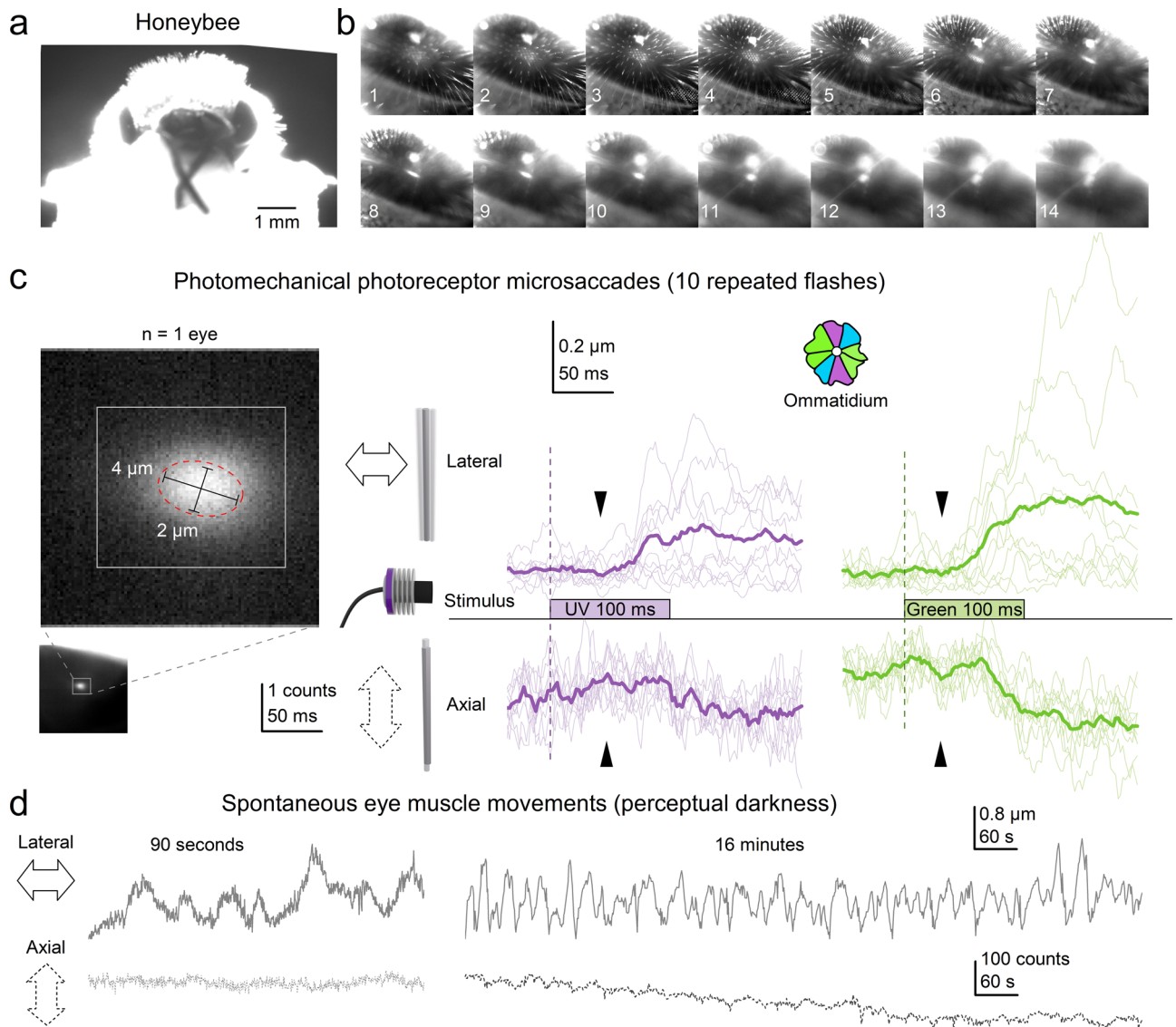

**Fig. 7 DPP microsaccades in honeybee fused rhabdom apposition eyes. a** Head-immobilised living honeybee prepared for the high-speed DPP imaging experiments. **b** Honeybee compound eye image series under antidromic infrared illumination, when the microscope gradually focusses inside the eye. First, the facet lenses appear (frames #3 and #4) before the DPP emerges as a small bright disk (frames #11 and #12). **c** High-speed imaging of photomechanical photoreceptor microsaccades. A characteristic slightly elongated DPP image of superimposed rhabdoms of neighbouring ommatidia. The DPP elongation is caused by the "ellipsoid" (non-spherical) honeybee eye shape. The scale bars give the actual rhabdom dimensions in ommatidia. The tested UV and green flashes evoked photomechanical photoreceptor microsaccades, seen as ultrafast lateral DPP "jump" and axial DPP darkening synchronised in time. The average DPP microsaccade to the green flash (green traces) was 1.54 times larger than to the UV flash (purple traces), consistent with the ommatidia having more 1.6 times more green photoreceptors (4) than UV (2.5) photoreceptors (inset). **d** Spontaneous slow eye-muscle-movement-induced (whole retina) DPP drifts were 10 to 100 times slower than light flash triggered photomechanical photoreceptor microsaccades. However, unlike in *Drosophila*[5], the sluggish eye-muscle-activity shifted the Honeybee DPP positions over seconds and minutes more than the ultrafast (<200 ms) DPP microsaccade displacements superimposed on them.

## Discussion

We recorded photoreceptor microsaccades across the wild-type *Drosophila*, *spam* mutant and honeybee compound eyes using a novel infrared GHS-DPP microscopy and analysed their active sampling kinematics. Remarkably, we found the *spam* mutants and honeybee generating ultrafast light-induced microsaccades akin to the wild-type *Drosophila*. Furthermore, in *spam*, the lateral microsaccade movements oriented locally, forming mirror-symmetric left and right eye sampling maps, largely similar to the wild-type flies. These results demonstrate that photoreceptor microsaccades are not limited to the open rhabdom eye design but also occur in fused rhabdom eyes. Most insects, including

honeybees, possess fused rhabdom eyes[22,23], in which photo-transduction reagents, including $PIP_2$, are thought to function alike in *Drosophila*[32,35]. Therefore, it seems probable that all insect compound eye photoreceptors would generate active sampling.

Photomechanical photoreceptor microsaccades are not reflex-like uniform contractions[4,5]. Instead, at each moment, they actively and continuously auto-regulate photon sampling dynamics by moving and narrowing the photoreceptors' receptive fields in respect to environmental light contrast changes to maximise information capture[4]. These dynamics rapidly adapt to immediate light history and are different at dim and bright

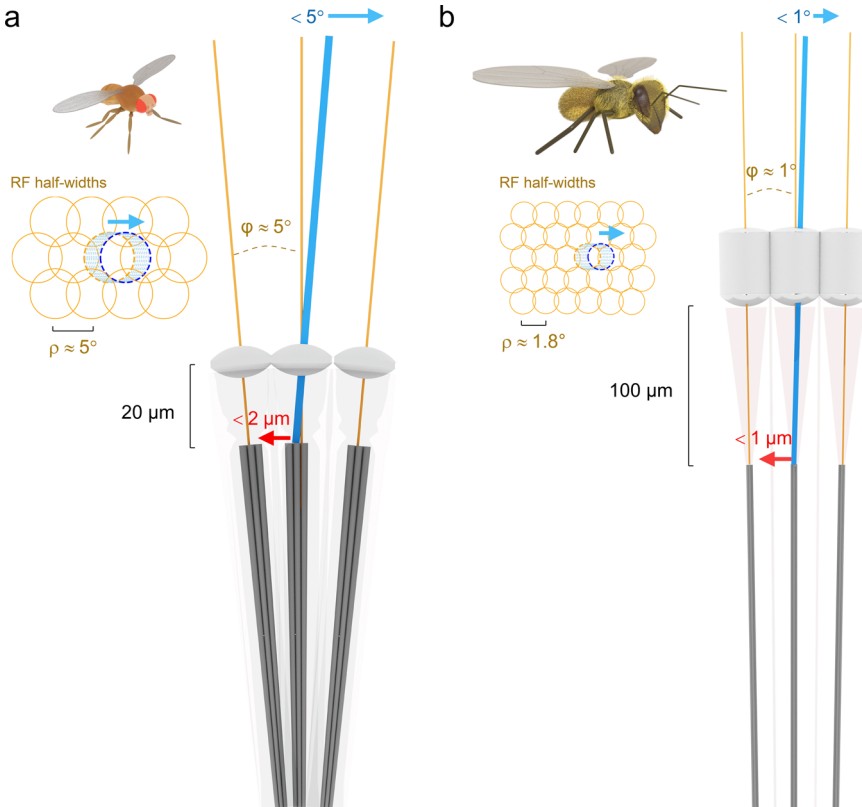

**Fig. 8 Fruit fly and Honeybee photoreceptor microsaccades scale with their receptive field half-width and interommatidial angle, presumably maximising retinal image acuity. a** *Drosophila* photomechanical photoreceptor microsaccades typically shift their rhabdomeres 1–1.5 µm laterally (max < 2 µm), equating to ~3–4.5° receptive field movements in the visual space. R1-R6 photoreceptors' receptive field half-widths (Δρ, retinal pixels) are between 4.5-6°, over-completely tiling up the visual space[4,5]. The average interommatidial angle[44] (φ) is 4.5–5°. Now consider this as image sampling by a digital camera. The spatial image information doubles when its sensor is moved, and two consecutive images are taken a 1/2-pixel apart and time-integrated for enhanced resolution. But if a pixel (receptive field) moves more, it eventually fuses with its neighbour (if that neighbour pixel was still – i.e. not detecting light changes). Because of this complete pixel fusion, acuity would decrease as the resulting neural image would contain fewer pixels. Therefore, by limiting micro-scanning to interommatidial angle, *Drosophila* can time-integrate a neural image, which greatly surpasses its compound eyes' optical limits[4,5].
**b** Honeybee photoreceptor microsaccades shift their receptive field maximally <1°, approaching the eye's average interommatidial angle[1]. Equally, such displacement is less than their average receptive field half-width (~1.8°) in the front of the eye[34]. As this active sampling strategy is broadly comparable to that of *Drosophila*, we predict that honeybee vision surpasses its compound eyes' static pixelation limit, similar to what we have shown for *Drosophila*[4,5].

conditions and to positive (light increments) and negative contrasts (light decrements)[4,5]. So, from the viewpoint of sampling theory, photoreceptor microsaccades are not passive but constitute a form of ultrafast *morphodynamic* active sampling[4,5]. However, in this study, we only examined photoreceptor microsaccades in one stimulus condition; to bright light flashes after prolonged dark adaptation.

The spam and honeybee DPP microsaccade displacements were generally smaller than the wild-type *Drosophila*'s. This finding is consistent with the inter-rhabdomeric coupling hypothesis[5]. The fused rhabdom rhabdomeres embrace each other and rigidify, and therefore, during microsaccades, they would have less flexibility to move sideways than the open wild-type *Drosophila* rhabdomeres. Conversely, in the much larger honeybee eye, the rhabdoms are further away from the ommatidium lenses, reducing their receptive field sizes and interommatidial angles[5,36]. The photoreceptor microsaccades then seem scaled down in proportion to the interommatidial angle, presumably for scanning the best image resolution (Fig. 8). These active sampling (or micro-scanning) strategies are not mutually exclusive. Both structure-function relationships could be evolutionarily tuned to scale the insect photoreceptors' active sampling dynamics to each species' unique visual needs. For example, we

would predict for fast-flying flies, such as houseflies (*Musca domestica*) and blowflies (*Calliphora vicina*), having more ommatidia tiling their eyes more densely, that their photoreceptor microsaccades be smaller and faster than those of slow-flying *Drosophila* of fewer less-densely-packed ommatidia. This way, a fast-eye's photoreceptor receptive fields would sample the world in higher velocity and resolution for higher visual information capture[37]—but these high-rate processes would make them metabolically more expensive[37]—than those of a slow eye. The fast eyes should also have more frontal ommatidia with the fastest microsaccades to accentuate acuity[4] and stereoscopic range[5] than the slow-eyes.

Ultrafast microsaccades of dissociated ex vivo *Drosophila* photoreceptors show both *lateral* and *axial* components[4], implying that underneath the ommatidial lenses, light changes make photoreceptors bounce inwards and outwards and sideways in a complex piston motion[4,5]. GHS-DPP microscopy can reveal both of these components (Figs. 5d, 7c), enabling us to estimate how they shape the way photoreceptors encode visual space in neural time through modelling[4,5]. The axial component pulls the rhabdomere away from the ommatidium lens to collect light from a narrower angle. The lateral component makes the resulting receptive field scan the visual space. GHS-DPP microscopy

produces 2D image sequences ideal for measuring the *lateral* photoreceptor microsaccade dynamics. However, the method detects less well their *axial* components, moving rhabdomeres to and from the ommatidial lens. The estimation becomes less reliable the further away the rhabdom/rhabdomeres are from the ommatidium lens. Following the laws of physics, the proportional DPP intensity change (brightening/darkening), indicating the axial rhabdomere movement, diminishes with distance[5]. Therefore, GHS-DPP imaging can underestimate the overall microsaccade dynamics if these show little sideways movement but have a larger (concealed) axial component. We suspect this would be the case with the honeybee photoreceptor microsaccades. Honeybee rhabdoms are 4.5 times longer than *Drosophila* rhabdomeres, thus having greater potential for axial contraction, but positioned about 100 μm from the ommatidium lens[38]; 5 times the distance in *Drosophila*[36].

The microsaccades in the *spam* mutants were slightly more rotated than the wild-type. Nevertheless, with such minor differences, their global maps should broadly match the forward flight optic flow field, similar to what we have previously shown for the wild-type[5]. Theoretically, this correspondence between active sampling and optic flow should improve the visual resolution of the moving world[5]. We further conjectured that since the microsaccades follow the R1–R2–R3 rhabdomere orientation axis[5] (Supplementary Video 6), their movement directions are set during development, perhaps guided by some lowest resistance (minimum energy) anchoring[5]. Thus, the observed wild-type and *spam* microsaccade direction differences could reflect slight differences in their R1-R7/8 rhabdomere orientations. Unfortunately, in the circular-symmetric *spam* DPP, the separate R1-R7/8 rhabdomeres are not directly identifiable under infrared illumination. In the future, this hypothesis could be tested by expressing GFP in selected R1-R7/8 photoreceptors or rhabdomeres. Moreover, because the high-speed imaging in the larger and densely pigmented honeybee eyes was experimentally challenging, we also left mapping their microsaccade movement directions for future studies.

Some sporadic interference in the measured microsaccades can originate from the intraocular muscles' activity, which exerts force on the retina. But since such muscle activity typically occurs over much longer time intervals[21] or during active viewing[6], being practically absent in firmly restrained flies[5] and sluggish in head-immobilised honeybees, the eye muscle induced movements had little influence on the ultrafast photomechanical microsaccades shown in this report (i.e. influencing them perhaps only through variable ommatidial tension).

However, during normal behaviours (in non-restricted, free-moving conditions), the local photoreceptor microsaccades and the global eye-muscle-induced whole retina movements must interact in active sampling. On the top of any eye-muscle-induced whole retina movements, the photomechanical photoreceptor microsaccades will ensue, leading to complex superimposed spatiotemporal ("super-saccadic") sampling dynamics. This sophistication arises because each retina movement will change its photoreceptors' light input, evoking their photomechanical microsaccades. Notably, active sampling can be even more elaborate if the eye-muscle-induced whole retina movements were partly voluntary and depended upon the attentive state. In those circumstances, an insect could use eye-muscle-induced retina movements together with other directional senses, such as antennal casting[39], to get a better idea of what an encountered object might be. After all, integration of multisensory information reduces uncertainty, increasing fitness.

Finally, we note that there are uncontrolled genetic differences between the wild-type and *spam* phenotypes, which could potentially contribute to their observed DPP microsaccade differences. Nevertheless, such differences would not alter the general demonstration of the active sampling of the fused rhabdom

eye photoreceptors as it occurs with somewhat comparable dynamics to the honeybee rhabdom.

To summarise, goniometric high-speed deep pseudopupil (GHS-DPP) microscopy provides an innovative non-invasive way to image photoreceptor microsaccades globally across the left and right eyes, or locally, in great detail at specific eye locations. We explained how to use it and gave out free open-source (GPLv3) software tools to quantify and compare active sampling in different insect eyes. Our results demonstrate active sampling both in open and fused rhabdom eyes. Thus, the GHS-DPP microscopy shows real potential as a powerful tool to study how the insect eyes actively sample the visual world.

## Methods

We describe the GHS-DPP imaging system hardware configuration, how to prepare the flies, experimental protocols, and data analysis tools and principles. All software and exemplary data[40]—from recording to analysis—are available under a free and open (GPLv3) software license in a GitHub repository: https://github.com/JuusolaLab/GHS-DPP_paper.

**Experimental setup**. The GHS-DPP imaging system's two primary components are a rotation stage system and a stereomicroscope (Fig. 2a). The rotation system allows precise control over the fly eyes' yaw and pitch using two perpendicularly mounted rotation stages (Thorlabs PR01/M, USA), mounted horizontally and vertically. An additional small 3-axis micromanipulator, connected to the vertical rotation stage, was used to control the fly's initial position. Crucially, the vertical rotation stage further rested on a 2-axis micromanipulator so that the intersection point of the two rotation stage axes could be centred at the microscope's field of view. The rotation stage positions were acquired digitally, using two 1024-step rotation encoders (YUMO E6B2-CW23E) and an Arduino board (Arduino Uno, Italy) running a custom C++ program (Fig. 2b). In addition, the rotation stages were fitted with stepper motors for fully automated experiments (Fig. 2a). Still, in this study, the experiments were performed manually, accurately focussing on DPP at all tested eye locations. Supplementary Video 1 shows how the GHS-DPP microscope system was put together. Note that the stereomicroscope's horizontal mounting is not critical but resulted from our earlier design choices. By having an upright rotation stage, this configuration works well for the binocular compound eye measurements.

The stereomicroscope was mounted sideways to function with the rotation stages. A high-intensity ultraviolet LED (UV-OptoLED, Cairn Research, UK) was inserted in the microscope's ocular slot, enabling direct light stimulation of the observed DPP rhabdomeres. This 365 nm UV-LED and the two infrared 850 nm LEDs (IR-OptoLED, Cairn Research, UK), which provided antidromic non-stimulating illumination of the DPP rhabdomeres (without activating their phototransduction) through the fly head capsule, were connected to their separate driver units (Dual OptoLED Power Supply, Cairn Research, UK). The LEDs were controlled over the BNC interface using a computer-connected data acquisition system (PCI-6221 with BNC-2090A and PCI-6733 with BNC-100, National Instruments, USA) (Fig. 2c). The two infrared LEDs were mounted apart from each other at different angles to prevent the pipette tip or the fly body from blocking the illumination, which would otherwise frequently happen with a single point source. Both the infrared LEDs were in bespoke holders, having a convex lens with adjustable lens-to-LED distance for beam focusing. The second unit mounted to the microscope was the high-speed optical camera (Orca Flash 4.0 C13440, Hamamatsu, Japan), which also sent a trigger signal over the BNC interface to the data acquisition system to time the stimulus delivery precisely. We typically acquired 2048 × 2048 pixel full-frame images at 100 fps, and occasionally—by cropping the sensor to 2000 × 200 pixel—collected images at 1000 fps. The camera had a transparent infrared and opaque UV filter on its pathway, ideally stopping the UV stimulation light from reaching and polluting the image sensor (Fig. 2d). Besides these filters, the microscope was configured with a beam-splitter (SZX-BS, Olympus, Japan), a photo adapter piece (SZ-PHA, Olympus, Japan) and a magnification changer (U-TVCAC, Olympus, Japan).

The setup was mounted on a vibration isolation table (air table), which uncoupled any building vibrations that could affect the motion analysis results. The rotation stage system was connected to the table by magnetic clamps (Magnetic Stand, World Precision Instruments), whilst the microscope system—attached to a thick steel pole of a steel base plate—was heavy enough to ensure its fixed position. The whole setup was enclosed inside a black-painted metal cubicle, with its only open side supporting black curtains for performing the experiments in controlled, dark conditions. Accordingly, the Arduino board's few surface-mounted LEDs were covered with electrical insulation tape to minimise any light noise.

**Flies and preparation**. We used wild-type *Drosophila* flies (Berlin) and a fused rhabdom *spam* null-mutant line (w; spam1/spam1 Frt; sqh-GFP/Tm6B, a gift from Andrew Zelhof) in the experiments. The flies were maintained in an incubator at 25 °C, under a 12:12 h light-dark cycle. Only healthy 3- to 20 days old male and

female flies, climbing up the vial, were selected for the experiments. We avoided using very young flies (<3 days) because their soft heads could bulge during the imaging, presumably due to spontaneous eye muscle activity. The flies were prepared for the experiments using the plastic pipette tip immobilisation technique[41] (Supplementary Fig. 1). Previously, we tested a copper hook tethering technique[5], allowing simultaneous behavioural experiments (Supplementary Fig. 1a). But because both these immobilisations gave broadly similar results[5], the more laborious and time-consuming tethering was not used here.

In the pipette-tip-fixation method, a 1000 μl plastic pipette tip was first linked to a funnel piece, and the funnel was connected to a vial full of flies. One fly at a time was lured into the pipette tip through the funnel driven by *Drosophila*'s innate geotaxis behaviour (Supplementary Fig. 1b). This way, there was no need to immobilise the flies by $CO_2$ or ice-cooling that could potentially affect the microsaccades. Next, the pipette tip was viewed under a stereo preparation microscope (Olympus SZX-9, Japan). At the same time, the fly was gently pushed towards the tip opening by puffing air from a hand-held syringe (Supplementary Fig. 1c). When needed, the pipette tip's opening was adjusted using a razor blade to ensure that the fly head passed through without any deformation, minimising structural damage to the eyes. The fly was air-puffed until its head and upper thorax protruded the tip end. When the fly was ideally positioned, it was quickly immobilised by applying melted beeswax on its thorax (Supplementary Fig. 1d) and ventrally of its head and proboscis (Supplementary Fig. 1e). In this correct position, the pipette would not shadow the dual-IR illumination during the experiments.

Additionally, the proboscis could have been pulled out with forceps and waxed on the pipette's outside, or the antennae waxed. However, these manoeuvres, which we often use in preparing *Drosophila* for intracellular electrophysiology[41,42], were omitted because they made no difference in the observed microsaccades. Furthermore, these procedures would have prolonged the preparation making, potentially increasing structural damage. Conversely, some more wax was routinely applied on the dorsal side of the head, under the ocelli, to secure the head position further. Finally, the pipette was cut from its large end using a razor blade (Supplementary Fig. 1f), and the preparation was carefully placed in the setup (Supplementary Video 2).

**Honeybee**. Honeybees (*Apis mellifera*) were kept indoors in a hive that provided the bees an outside (nature) access through a plastic portal. Worker bees were captured from the portal and prepared using the pipette tip technique similar to the *Drosophila* with few adjustments. First, the pipette tip's small opening was cut larger to fit the bee head. Second, the bee was ice-cooled before the waxing to prevent it from escaping. During the cooling, the bee inside a plastic vial was placed in ice until its leg movements temporarily halted. Although the bees quickly recovered, as judged by their antennal movements, the ice-cooling may have generally affected their physiology. Finally, a cuticle section (between the antennae and ocelli and the dorsal part of the eyes) was removed by a razor blade after the waxing. This microsurgery greatly increased the DPP image's brightness, allowing us to image photoreceptor microsaccades in high frame rates (100 Hz and 500 Hz).

**Data acquisition and software**. To enable non-specialist users to operate the GHS-DPP microsaccade imaging experiments, we created a free and open (GPLv3) recording software called *Gonio Imsoft*[43]. *Gonio Imsoft* interfaced with the open-source microscopy software *MicroManager* to control the high-speed camera. In addition, it used the *NI-DAQmx* module (controlling the data acquisition) and the *PySerial* module (communicating with the Arduino microcontroller), reading out the rotary encoders. *Gonio Imsoft* ran on a Windows 10 platform.

We used the Olympus DF-PLAPO 1X objective for typical experiments with the microscope's continuous zoom maxed out and the additional magnification changer set to the 2x-position. In this configuration, a small-to-average-size fly head falls entirely within the field of view. The microscope's light path selector was set to its mid-state, resulting in an 80%/20% light intensity split between the camera and the eyepieces, respectively. The two infrared illumination LEDs were focused on the back of the fly head. Using the 2-axis micrometre, mounted on the horizontal rotation stage, the vertical rotation stage was centred so that the fly head, when brought to the field of view by the 3-axis micromanipulator (mounted on the vertical rotation stage), remained in the centre of the field of view in all possible horizontal and vertical rotation combinations (Supplementary Video 2). However, the microscope's focus remained only approximately correct since the fly head is not perfectly round and hence refocusing was needed to maintain a sharp DPP image during the experiments.

The microscope was first focused so that the ommatidial lenses appeared crisp and clear. Then, the focus was brought deeper into the eye until the DPP became clear and visible. In this ommatidial axes' converge point, images formed by individual ommatidia superimpose, creating a magnified, virtual image of the rhabdomere tips[11] (Supplementary Video 4). Using our computer graphics (CG) simulations on wild-type and *spam* eyes, we further confirmed that the retinal patterning (Fig. 1a, b) indeed resulted in a one-spot DDP image in the *spam* (Fig. 1c).

The DPP is purely an optical phenomenon, informing us more about how the observed rhabdomere summation image is affected by the used microscope system[11] than how the fly compound eye converges visual inputs from the world[28].

Therefore, the DPP cannot tell us too much about the eyes' neural superposition design, in which R1-R6 photoreceptor outputs from six neighbouring ommatidia, sampling light over the same small visual area, are synaptically pooled. Nevertheless, it is helpful to estimate the number of ommatidia forming in the optical DPP image as this improves our understanding of any blurring and irregularities in it. On a hexagonal tiling, the number of hexagons within an $n$th concentric hexagonal circle (Fig. 9a, b) can be calculated using the formula

$$
\begin{cases}
N = 3n(n-1)+1 & \text{if } n \geq 1 \\
N = n & \text{if } n < 1 \\
N = 0 & \text{if } n \leq 0
\end{cases}
\tag{1}
$$

The values of $N$ are known as the centred hexagonal numbers. The piecewise formulation also accounts for the $n < 1$ cases, usually left undefined. By finding the right $n$, we use it to denote the number of ommatidia participating in the DPP image formation. From the eye-microscope geometry (Fig. 9c, d), we can estimate the value of $n$, after which the illumination light rays no longer can enter the microscope's entrance pupil, and write

$$
n = \frac{\theta + 2(0.5 - c)\alpha}{\triangle \varphi} + 1
\tag{2}
$$

$\theta$ is the half-angle subtended by the microscope lens to the eye, $\triangle \varphi$ is the interommatidial angle, $c$ is the minimum counted contribution term and $\alpha$ is the mean deviation angle between the outermost rhabdomere receptive axes and the ommatidial axis. If we were to estimate the number of ommatidia that project all their rhabdomeric light to the microscope, $c = 1$, and conversely, if we were interested in ommatidia casting *any* rhabdomeric light to the microscope, $c = 0$. For counting ommatidia with other contributions, the correct $c$ depends on the microscope NA. But for a 50% minimum contribution estimate, we can use the former estimates. Then next, ideally, if the rhabdomeres formed a single point on the ommatidial axis (and the eye was perfectly spherical), $\alpha$ would be zero. And, if the rhabdomeres were organised into a hexagonal shape so that the neighbouring rhabdomeres' receptive fields (RFs) perfectly overlap at infinity, $\alpha$ would nearly equal the interommatidial angle $\triangle \varphi$.

By replacing the half-angle with the numerical aperture (NA) and considering that in *Drosophila* 1 μm displacement results in a 3° angular change[5], we can write

$$
n = \frac{\sin^{-1}\left(\frac{NA}{m}\right) + 2(0.5 - c)(3°/\mu m \, r)}{\triangle \varphi} + 1
\tag{3}
$$

$NA$ is the numerical aperture of the used microscope, $m$ is the refractive index of microscope objective immersion medium ($m = 1$ for air), and $r$ is the radius of a circle in the ommatidial retinal plane that representatively contains all the rhabdomeres. The numerical aperture in our microscope system was 0.11, and the *Drosophila* interommatidial angle is approximately 5°[36,44]. Since the DPP image is a ~10× magnified image of the rhabdomere tips, the $r$ parameters for the wild-type and *spam* can be estimated directly from the DPP still images as 2.8 μm and 1.5 μm (Fig. 9f). Inserting these values in (3) and then calculating (1), we approximate that in our microscope system, 18 ommatidia would contribute at least 50% to the deep pseudopupil image formation in the wild-type and 12 ommatidia in the *spam* eye. For one rhabdomere in the wild-type DPP pattern, we can similarly estimate that its $r$ is 0.78 μm (Fig. 9f), suggesting that 10 ommatidia form it. Notice, however, that because of the structural asymmetricities, these ommatidium counts are likely overestimated. For example, the *Drosophila* ommatidial R1-R6 rhabdomeres are not hexagonally arranged around the lens centre. Instead, their sizes and distances vary[4], forming an asymmetric (slanted) trapezoid arrangement (Fig. 3c), and the eye is not perfectly spherical. Moreover, infrared illumination, by passing through the ommatidial screening pigments that block non-axial green and UV transmission, could potentially merge more ommatidia into the DPP image than visible light. Nevertheless, since these upper-bound estimates vary with the used microscope systems (Fig. 9g), we calculated them for a range of air-objective NAs (Supplementary Table 1).

In this study, we performed two kinds of experiments: *local recordings* at a fixed position on the left and right eyes and *global recordings* across the eyes. In the local recordings, we imaged one location on the left eye (+28° horizontal and −37° vertical) and another on the right eye (−28° horizontal and −37° vertical) because, at these locations, the upper infrared LED illuminated the eyes ideally, forming the crispest DPP. The stimulus UV-LED was flashed for 200 ms while simultaneously the high-speed camera acquired images 100 frames per second, yielding 20 image frames per flash. As a standard practice, this recording process was repeated 25 times to obtain the mean photoreceptor microsaccade estimates and inspect their variability in an individual fly. Between the repeats, we initially used 10 s inter-stimulus-interval (ISI) for both the wild-type and spam flies. But ISI was later shortened ISI to 2 s, as there was no significant reduction in the response amplitude. All images were saved as 16-bit unsigned-integer, grayscale TIFF images.

The global imaging procedure was similar, but it was performed only once at each location. Instead, after each flash, the fly was rotated in 10° steps from −40° to +50° (limited by the vertical rotation stage, covering the microscope objective or the two illumination LEDs) using the horizontal rotation stage. After completing the horizontal "line scan", the vertical rotation stage was advanced in a 10° step,

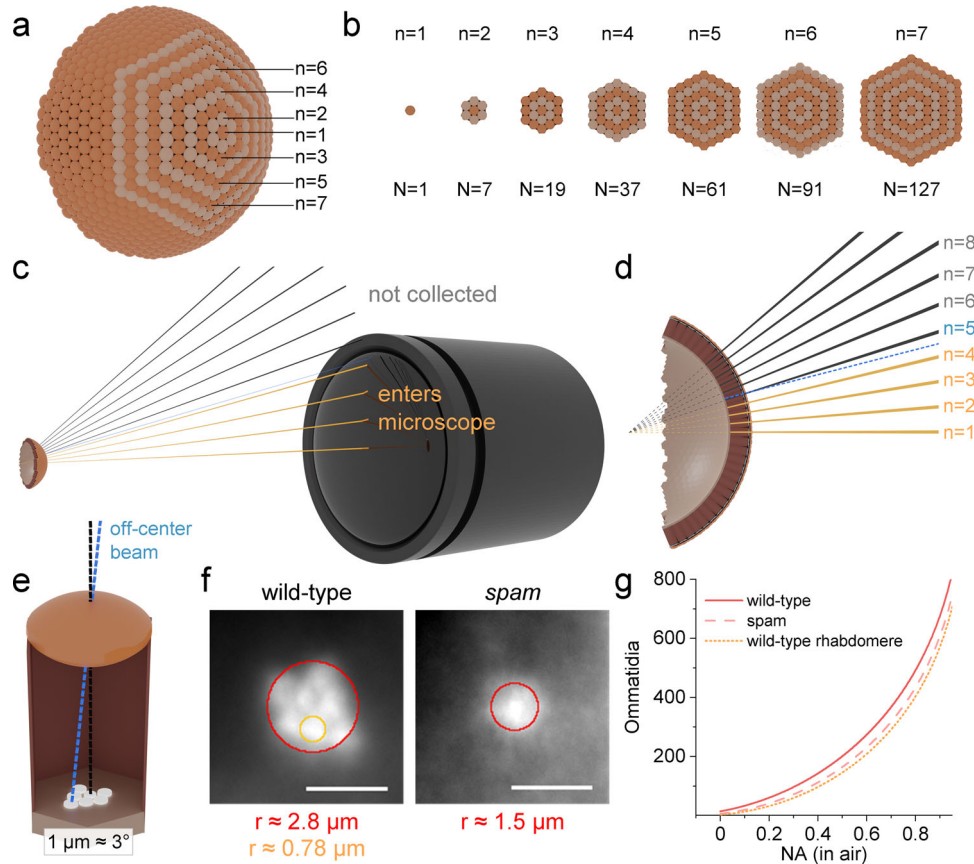

**Fig. 9 The amount of DPP forming ommatidia depends on the microscope numerical aperture. a** A spherical compound eye illustrated with seven centred hexagonal rings. **b** The $n$th centred hexagonal number quantifies the number of ommatidia, $N$, within the $n$th ring. **c** Light rays emerging from the retina midpoints can be collected by the microscope only if their incident angle is smaller than the microscope's half-collection angle, $\Theta$. **d** Some light from the uncollected ommatidia may reach the microscope if it originates off-centre (blue). **e** light originating 1 μm off-centre is approximately 3° diverted from the optic axis. **f** The $r$ parameters were estimated from the wild-type and *spam* DPPs, and one wild-type DPP rhabdomere. The scale bar corresponds to 50 μm in the DPP images and 5 μm physically in the retina. **g** The amount of DPP forming ommatidia (here $c=0$) rises sharply with the increasing microscope numerical aperture, NA.

covering a range from −110° to +110°. By rotating the fly and imaging in between, we scanned approximately 200 distinct locations on the left and right eyes. Overall, it took about 10 s to reorient the fly, refocus at the deep pseudopupil and start the image acquisition protocol again.

**Data analysis.** To quantify the three-dimensional rhabdomere movement fields, we created a free and open (GPLv3) data analysis software called *Gonio Analysis*. It allows drawing regions of interest (ROIs) around the deep pseudopupil, performing motion analysis, and finally, translating the motion results from the camera image coordinates to the fly's 3D frame-of-reference, using the digitally read rotation stage values. Here, we shortly describe its data analysis principles.

Rectangular ROIs were drawn by hand for the first frame of each location only by selecting the deep pseudopupil. Next, these pseudopupil images were used as template images for cross-correlation based motion analysis (Fig. 10a). We used the computer vision library *OpenCV*[45] and its *matchTemplate* routine for the following 2D cross-correlation

$$R(x,y) = \frac{\sum_{x',y'}(T'(x',y')\,I'(x+x',\,y+y'))}{\sqrt{\sum_{x',y'}T'(x',y')^2 \sum_{x',y'}I'(x+x',\,y+y')^2}} \quad (4)$$

$$T'(x',y') = T(x',y') - \frac{1}{w*h}*\sum_{x',y'}T(x'',y'') \quad (5)$$

$$I'(x+x',\,y+y') = I(x+x',\,y+y') - \frac{1}{(w*h)}*\sum_{x'',y''}I(x+x'',\,y+y'') \quad (6)$$

$R$ is the two-dimensional cross-correlation image, and $R(x,y)$ is its value at the pixel $(x,y)$. $x'$, $x''$ and $y'$, $y''$ are summation indices within ranges [0, 1, 2, …, $w−1$]

and [0, 1, 2,…, $h−1$], $w$ and $h$ are the width and height of the template image. $I$ is the source image, and $T$ is the template image.

In the cross-correlation results of the images $R$ produced by the template matching (Fig. 10b), the higher values were signed for the higher similarity between the template and source images at each location. Therefore, using two source images acquired at different times, the template displacement between these images can be calculated by comparing their resulting cross-correlation image's peak values (Fig. 10c). Furthermore, we visually confirmed some motion analysis results by creating videos in which the rectangular ROI box was moved according to the motion analysis results, readily following the moving DPP. On the other hand, we note that computing the complete cross-correlation with uncropped source images (cf. Fig. 1) is inefficient and can lead to a false match. Therefore, we instead performed the cross-correlation only in the near vicinity (30 pixels) of the DPP cropping without truncating the responses.

In the template matching, the motion analysis results were given in the camera image coordinates. To translate them from the camera system to the 3D space in the fly's frame of reference (Fig. 10d), we used the digitally read vertical $v$ and horizontal $h$ rotation stage values on the following set of equations

$$\begin{cases} y = \cos(h)\,\cos(v) \\ z = y\,\tan(v) \\ |x| = \sqrt{1-y^2-z^2} \end{cases} \quad (7)$$

to calculate the microscope's $(x, y, z)$ location. Using the same equation set, in short, $P(h, v) \rightarrow (x, y, z)$, we then calculated the camera x unit vector as the vector from $P(h, v)$ to a slightly displaced point $P(h + \triangle h, v)$

$$\hat{i}_{cam}(h,v) = \overrightarrow{P_{(h,v)}\,P_{(h+\triangle h,v)}} \quad (8)$$

where $\triangle h$ is ideally as small as possible but large enough not to cause errors because of limited floating-point precision. Since the y-camera-unit only depends

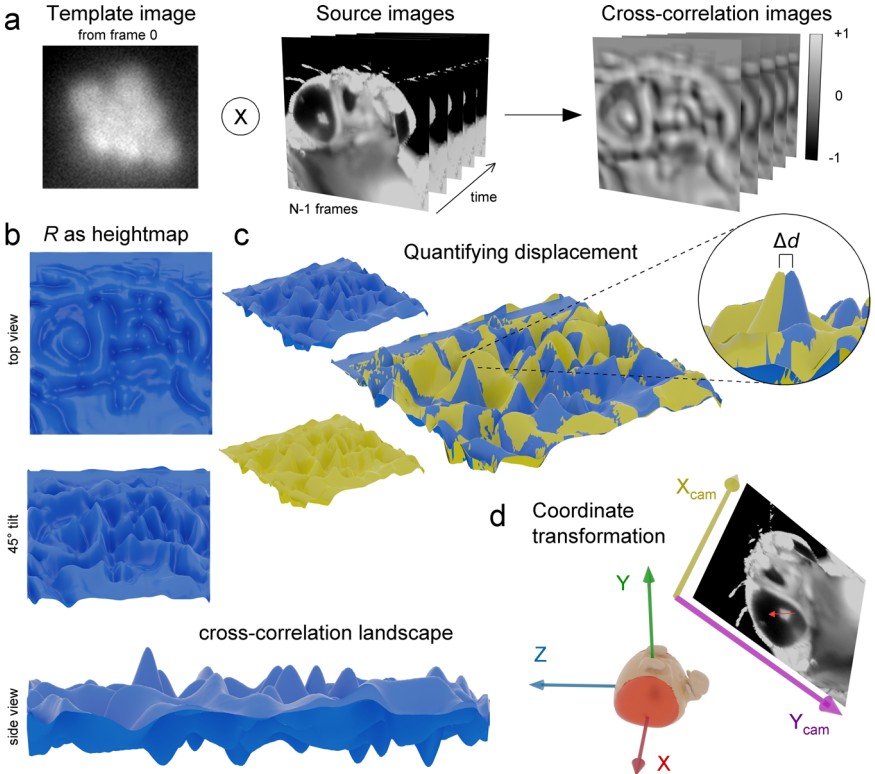

**Fig. 10 Cross-correlation based motion analysis by template matching. a** The first frame of the DPP image, manually cropped, was used as the template image. By 2D cross-correlation analysis, the DPP image locations were searched among the source images. **b** A cross-correlation result image R can be presented as a height-map, giving a better view of the cross-correlation landscape. **c** Two superimposed R height-maps reveal a small shift Δd in the location of the primary maxima; This is the DDP displacement. **d** The motion analysis results were transformed from the camera image coordinates to the fly coordinate system using the rotation stage values.

on the vertical rotation and its x-component in the fly coordinate system is conveniently zero (i.e. the y-unit-vector is always perpendicular to the great circle arc that the microscope travels along from the fly's point of view), it is simply

$$\hat{j}_{cam}(v) = -\sin(v)\hat{j} + \cos(v)\hat{k} \qquad (9)$$

Finally, the $(x, y, z)$ movement vectors can be then calculated using camera unit vectors as

$$\boldsymbol{v}\left(m_x, m_y, h, v\right) = m_x\hat{i}_{cam} + m_y\hat{j}_{cam} \qquad (10)$$

where $m_x$ and $m_y$ are the camera image $x$ and $y$ movement values produced by the cross-correlation.

We averaged the results over many flies using simple $N$-nearest neighbour interpolation for the microsaccade vector maps acquired in the global recordings. For each interpolation point, from each of the N imaged flies, the nearest 3D vector was selected, but only if the angular distance of the 3D vector was not larger than 2 times the angular interpolation step of 5°. And, these equal or less than $N$ vectors were averaged together only if there were $N/2$ of them or more. The difference in the wild-type and *spam* vector maps were calculated point-wise as

$$e\left(\boldsymbol{v}_{\mathbf{wtb}}, \boldsymbol{v}_{\mathbf{spam}}\right) = \frac{1}{\pi}\cos^{-1}\left(\frac{\boldsymbol{v}_{\mathbf{wtb}} \cdot \boldsymbol{v}_{\mathbf{spam}}}{||\boldsymbol{v}_{\mathbf{wtb}}|| \, ||\boldsymbol{v}_{\mathbf{spam}}||}\right) \qquad (11)$$

where $\boldsymbol{v}_{\mathbf{wtb}}$ and $\boldsymbol{v}_{\mathbf{spam}}$ are the microsaccade vectors and the operators $\cdot$ and $|| \, ||$ denote the inner product and the vector norm, respectively. Finally, for rotation direction analysis, we rotated the vectors on the x-axis and calculated whether the spam vectors were rotated clockwise or counterclockwise compared to the wild-type and then used this result to sign the error in (11).

In the local recordings, we focused solely on the directionless microsaccade magnitude, calculated using the Pythagorean theorem from the camera coordinate movement values. From these animal-specific mean magnitude traces, we calculated the microsaccade amplitude, speed, logistic growth factor and half-rise time quantifications. The probability graphs were calculated as 1D-histograms at each time point, with 20 stitched together to cover the whole imaging period. The

total displacement values were quantified from the magnitude data by simply taking the mean of the last 7 data points, the last 70 ms of the imaging. The maximum speed was calculated from the highest value between frames displacement. For the logistic growth factor and the half-rise time, we fitted the data with the sigmoidal logistic function

$$f(t) = \frac{L}{1 + e^{-k(t-t_0)}} \qquad (12)$$

where $L$ is the maximum value corresponding to the total rhabdomeric displacement, $k$ is the logistic growth factor that in our case characterises the microsaccade activation phase time-duration, and $t_0$ is the half-rise time.

**Statistics and reproducibility**. The presented results are readily reproducible. Every healthy *Drosophila* with functional vision will show them. The figures and figure legends give the sample sizes (how many flies were used) and the number of recorded responses (to repeated stimulation). In contrast, because honeybee has black "armoured" head cuticle, which makes preparing them considerably more complicated, we show only exemplary (repeatable and reproducible) results from one bee. These results reveal photoreceptor microsaccades occurring also in the bee eye, with comparable ultrafast dynamics to *Drosophila*, and demonstrate that GHS-DPP microscopy can capture them too. All the quantified local recording parameters (displacement, speed, growth factor and rise-time) appeared reasonably normally distributed. However, since the *Drosophila* wild-type sample size was 3 times larger than the *spam* one, we used Welch's $t$ test[46] that performs better with uneven samples for all comparisons.

**Reporting summary**. Further information on research design is available in the Nature Research Reporting Summary linked to this article.

## Data availability

Datasets for this study can be found in the https://github.com/JuusolaLab/GHS-DPP_paper

## Code availability

The software code for this study can be found in the https://github.com/JuusolaLab/GHS-DPP_paper

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

## Acknowledgements

The authors kindly thank Andrew Zelhof for the spam fly line and Gonzalo de Polavieja, Roger C Hardie, Merlin Juusola, Gregor Belušič and the Juusola laboratory members for discussions and comments. This work was supported by Jane and Aatos Erkko Foundation Fellowships (M.J. and J.T.), The Leverhulme Trust (RPG-2012–567: M.J.), the BBSRC (BB/F012071/1, BB/D001900/1 and BB/H013849/1: M.J.), the EPSRC (EP/P006094/1: M.J.), the Open Research Fund of the State Key Laboratory of Cognitive Neuroscience and Learning (M.J.), High-End Foreign Expert Grant by Chinese Government (GDT20151100004: M.J.).

## Author contributions

Conceptualisation: M.J., J.K., J.T.; Data curation: J.K., N.M.; Formal Analysis: J.K.; Funding acquisition: M.J.; Investigation: J.K., N.M., J.T., M.J.; Methodology: J.K., J.T., M.J.; Project administration: M.J.; Resources: M.J.; Software: J.K., J.T.; Supervision: M.J.; Validation: J.K., J.T., M.J.; Visualisation: J.K., M.J.; Writing—main paper original draft: J.K.; Writing—review and editing: M.J., J.K., J.T.

## Competing interests

The authors declare no competing interests.
