## [Peer Review File · Communications Biology]

Reviewers' comments:

Reviewer #1 (Remarks to the Author):

BRIEF SUMMARY

This is a fascinating and comprehensive description of the approach and methods required to directly observe a new phenomenon in insect vision: photoreceptor microsaccades - the light-activated movement of the photoreceptor complex (rhabdomere) under a single lens of the compound eye. These photoreceptor microsaccades would be important to 'actively sampling' the visual scene in response to light increments. This, in turn, would enhance spatial resolution of the neural image formed by the brain. So now the insect compound eye is not a fixed sampling structure, but rather shows biomechanical responses across the retina that may be highly adaptive for spatial vision. The manuscript is fairly well written, although can be unnecessarily verbose and confusing in some sections. The approach is well detailed and documented. There are, however, numerous opportunities for improving the manuscript to enhance its readability and impact.

OVERALL IMPRESSION

My overall impression was that this is excellent: I want to build this system and observe these phenomena myself.

SPECIFIC COMMENTS

CONCEPTUAL FRAMEWORK

The term "photoreceptor microsaccades" needs to be better distinguished from "microsaccades" as they are typically understood. Here, if I am reading this right, each ommatidium acts independently, responding to illumination locally at its lens, and producing a mechanical movement in the single direction specified morphologically by the axis of the rhabdomere. The global vector map of local PMS shows a 'flow field', but this field is not co-activated as a population. Rather, if you were to observe the full array of PMS in response to the movement of a natural scene, the PMS are activated by the spatially localized light levels. This is very different from retinal microsaccades, in which the entire retinal array is refreshed by the displacement of the full epithelium. For that matter, the term 'microsaccades' was meant to distinguish very small amplitude eye movements from much larger amplitude 'saccades' that reorient gaze. But here there are only 'photoreceptor saccades'...they are always 'micro'. Finally, saccades as they are traditionally referred to are unitary fixed action patterns. But here, the movements you characterize vary widely in amplitude. Are they really 'saccadic' in that sense?

DETAILS (in approximate order of appearance in the text)

Microsaccade vector field movies: the blue is too low contrast against a black background. Similarly, the blue boxes on the microsaccade movie is very hard to see against the dark background, also in the static figures.

In movies as well as figure panels: please add egocentric eye axes: frontal-backward, dorsal-ventral, otherwise it is very hard to orient. For example, in your movie, what does "Dorsal view" mean? Are we viewing the vector field from the dorsal perspective, in which case frontal (anterior) is upward? For Anterior view, is dorsal upward? Label these everywhere.

Make clear in the text that you are stimulating the WHOLE EYE with UV light. In which case, what is the difference between "global" and "local" experiments?

WOW! Note the highly prominent antennal movements in your photoreceptor microsaccade movie! Cite paper from Dickinson lab (J. Neuroscience?) showing optomotor antennal movements - are these phenomena related? Discuss!

Do other wavelengths of light activate photoreceptor microsaccades?

L42: ...shift one eye's sampling matrix globally relative to...

Introduction: you refer to the photoreceptor saccades as being a local phenomenon, whereas

intraocular muscles act globally. Do you know for sure that each rhabdomere acts independently? The field of illumination is large (this point also needs to be clarified within the narrative), so although you only image individual rhabdomeres via DPP, others may be co-activated, right? Or are you speculating that the spatial distribution of photoreceptor microsaccades mimics the local light stimuli generated by the spatial arrangement of light-dark edges moving across the eye?

Figure 1: (a,b) indicate (estimate) the imaging plane on the vertical ommatidia drawings.

Figure 1: You need to show raw data, raw images. It is fine to also show your CG modeled images alongside, but not in place of. Similarly, L63: "virtual images" are inappropriate for a methods paper, at least as the narrative stands in its current form. Most readers will become cross with this approach.

Figure 2: Is there a particular reason that the stereoscope is mounted horizontally?

Why do you choose UV wavelength to activate photoreceptor microsaccades (PMS)? Do other wavelengths activate the PMS? I realize that you have another paper under review, posted to bioRxiv, which I read through in order to better understand this one. But this paper has to 'stand alone' and so there needs to be a bit more conceptual unpacking, I think. Also, does full-field illumination activate ALL of the PMS together? Does screening pigment localize them, particularly for shorter wavelength light? Maybe some of these issues are best treated in the results-focused companion to this method-focused one.

Figure 3: please re-emphasize that the DPP that you image is a SINGLE rhabdomere from a SINGLE ommatidium, but that the imaging optics magnify it to occupy many more than one ommatidium in the image. Do I have this right? Please make this clear in the text.

Figure 3a-e: blue rectangle is hard to see against dark image

Figure 3c: perhaps show ONE spam DPP image?

Figure 3d caption: missing word "...appear as..."

Figure 4: the supplement video showing photoreceptor microsaccades is absolutely convincing. Yet, some readers will not wish to look at the supplement (I know....) so I would suggest a temporal sequence of single frames showing a photoreceptor microsaccade.

Figure 4c,d: indicate D-V, A-P axes

L126: This point, that the axis of the photoreceptor microsaccade aligns with the rhabdomere axis is crucially important as it speaks to mechanism. You should show a superposition of Fig 3c and 4c to show this.

SPAM mutant: why don't you simply test an insect with an apposition eye instead of the mutant *Drosophila* with a fused rhabdom? Do you expect that a fused rhabdom eye would have more or less disruption of the photoreceptor microsaccadic orientation map?

L133 extra word "the"

L136: the use of 'global' vs 'local' is very confusing here. You could clear this up in the narrative. "In order to obtain more robust results, we re-sampled photoreceptor microsaccades 25 times from a single ommatidium, located at ____." Also at L151 "We characterized variation across individual flies by re-sampling from a single retinal location..." This issue comes up again on L180; in fact ALL of your measurements are "local". You only image ONE DPP at a time.

L182-183: I do not follow the premise on the lever action here. Can you elaborate a bit?

Here, this is a methodology paper, in which you do not present evidence for "active sampling". In fact, one could conclude from the data presented here that the DPP microsaccades are reflexive,

particularly since they are always accompanied by antennal movements in the video. I wonder whether it makes sense to revise this to de-emphasize the interpretation of active sampling, and that the DPP movements are 'microsaccades' at all. You merely present evidence for light activated DPP movements.

L199 extra word "how"

Reviewer #2 (Remarks to the Author):

This manuscript describes a novel technique to measure photomechanical microsaccades in compound eyes. The authors utilize high-speed imaging to characterize the contraction of the photoreceptors to flashes of light in both open and fused rhabdom eyes. This approach enables the authors to investigate how these contractions differ between the two eye types and highlight an additional strategy by which compound eyes actively sample a visual scene in a way that resonates with the natural visual flow.

This study is interesting and introduces a technique that enables further investigations in active vision not only through self-motion but through a mechanism that is intrinsic to the compound eye. The perceptual benefits of such contractions have been reported previously by this group (Juusola et al, 2017, eLife); and the results here provide a way to investigate in much greater resolution the characteristics of both the global and local motion patterns. The methods are thoroughly described, and software approaches are openly available. Overall, the results are sound, and I look forward to the future exciting results that such an approach can provide to the field of active vision.

Specific Comments:

1. Line 80: "bellow" should be "below"
2. Figure 4 Caption: "d, Mean microsaccade vector map of spam -flies-"
3. Figure 4e-f and lines 119-123: It is unclear what "absolute difference" as collinear and perpendicular is quantifying, and how it is different from the directional difference. Please elaborate or include a definition in methods.
4. Line 199: "Both of these components shape the way -x-how-x- a photoreceptor encodes ..."
5. Figure 5: "f-j" should be "f-i"
6. The speculation and discussion following the analysis of variability of repeated responses and variance of displacement (Fig. 5j-k) are difficult to follow. How does such variation indicate purposeful regulation of these contractions?

Reviewer #3 (Remarks to the Author):

This paper describes a technique to measure lateral photoreceptor movements during light stimulation in vivo for *Drosophila*, and potentially other compound eyes. The authors combine deep pseudopupil imaging (with infrared, non-stimulating light), high-speed cameras, and a goniometer to quantify the photoreceptor microsaccades that result from membrane contraction during light absorption. They compare rhabdomere movements in wild-type *Drosophila melanogaster*, with movements in a mutant possessing a fused rhabdom, similar to typical apposition eyes.

Major:

A big conclusion of this paper is that "photoreceptor microsaccades are not limited to the open rhabdom eye design but can also occur in fused rhabdom eyes", such as honey bees, and the vast array of apposition eyes that aren't neural superposition. However, a mutant fruit fly certainly doesn't demonstrate this convincingly, as it may display this ancestral trait (a fused rhabdom) in a very different way from species that develop this way naturally (without a mutation). This is like claiming the presence of eyes on a blind mutant shows that eyes occur in other animals that don't have vision. The model system doesn't have quite this much power---even if it is likely to be true,

I need to see it in at least one animal with proper fused rhabdoms to believe this claim.

A figure showing the light paths for deep pseudopupil imaging would be quite helpful for most readers trying to follow the technique. Figure 1 shows DPP images nicely, but it could easily also incorporate a ray trace illustration, which would help readers interpret, for example, the central left and right images in Figure 3c, which are otherwise confusing. Figure S2 has something like this, or something like 7.11 in *Animal Eyes*. It would help readers understand, for example why "to observe a well-defined, clear DPP pattern in dipteran eyes requires precisely 367 organised rhabdomeres across the neighbouring ommatidia (ln 66)"

The paper also lacks a good image showing movement of the rhabdomeres. They are characterized in figure 4, but I was surprised by the lack of a clear figure showing the rhabdomeres before and during the UV light flash, to illustrate where the vectors in 4c and 4d actually come from.

An easy issue to address is that the many figure panels are far too small, with axis labels and arrows much smaller than the body text, and severely pixelated when I zoom in to see better. In 3b, 3e, 4b, and 5b I can barely make out the DPP images, which ought to be the highlight of this paper.

Minor:

In 25 "These results show different ways how vision converts space into time"

-I understand what you are getting at, that spatial light patterns in the environment become temporal electrical signals of a photoreceptor, but "converts space into time" is too compact, and makes it sound like a physics paper on relativity.

In 48 "atomic force microscopy (AMF)"

-AMF or AFM?

In 107 figure 4.

-4b on the right appears to show the microsaccade movement, but this isn't described in the legend, only the rotation relating to the right image. Are the top and bottom right images wt and spam?

-4e and 4f compare the absolute and directional difference of vector maps, 4c and 4d. I would have thought this would be a point by point difference in vector length for 4e and difference in angle for 4f. But the scale bar in 4e is labeled 'perpendicular' to 'colinear', which sounds like a directional difference to me. I need clarification here.

In 125 "These results demonstrate active sampling - by photomechanical photoreceptor microsaccades - happening in fused rhabdom eyes."

-How do the results demonstrate active sampling? They demonstrate motion in response to light, but this may, in principle, be non-functional in the spam mutant, just a vestigial response that does not affect, or maybe even reduces sampling in the fused rhabdom.

In 134 figure 5

-'Displacement' is usually a term for a vector, but except for the images in 5c which show an arrow, the authors are using it as a scalar (5f, 5k, legend), the magnitude of displacement. Then in the main text they use the term 'displacement amplitudes', seeming to indicate a vector again.

This should be made consistent, and if the authors decide not to use 'displacement' as a vector, explain early on in the text.

Editors and Reviewers,

We thank the reviewers for useful feedback, which has helped us improve this paper. We have now considered their suggestions and carefully edited the paper to make its findings clear and accessible for a broad range of readers.

Below, **our replies** to each Reviewer's specific comment (the numbering is ours) are shown in **blue**. In a separate pdf file, the corresponding changes made in the paper are shown in **red**.

We have also edited the manuscript and its figures to be consistent with the *Communications Biology* house style.

We hope that with these changes, this manuscript is now suitable for publication in *Communications Biology*.

Dear Professor Juusola,

Your manuscript entitled "High-speed imaging of light-induced photoreceptor microsaccades in compound eyes" has now been seen by 3 referees, whose comments are appended below. You will see from their comments copied below that while they find your work of potential interest, they have raised quite substantial concerns that must be addressed. In light of these comments, we cannot accept the manuscript for publication, but would be interested in considering a revised version that addresses these serious concerns.

We hope you will find the referees' comments useful as you decide how to proceed. Should further experimental data or analysis allow you to address these criticisms, we would be happy to look at a substantially revised manuscript. However, please bear in mind that we will be reluctant to approach the referees again in the absence of major revisions.

In particular, please note that the following revisions would be necessary for us to contact our referees again:

1. Especially, R1 raised two major concerns: (i) Clarification and distinction between "photoreceptor microsaccades" vs regular microsaccades. (ii) UV light stimulation vs other wavelengths. R3 is concerned about the conclusion that (iii) "fused rhabdom insect eyes show lateral microsaccades" derived from the mutant *Drosophila* experiment only, instead of non-mutant. (iv) R3 also would like to see more details of the optical system. Please properly address all the comments/suggestions raised by the three reviewers and submit a revised manuscript for further considerations.

1. Our reply:

(i) We now have added a new **Fig. 1(a)** to show what the ultrafast photoreceptor microsaccades are. In this new figure and the Introduction, we now explain in detail that the microsaccades are local ultrafast photomechanical photoreceptor contractions, evoked by spatially-localised light intensity differences in the visual environment. In addition to photoreceptor microsaccades, intraocular muscles can shift the whole retina within the strict structural limits of the left or right eyes. However, these movements in immobilised *Drosophila* and honeybee are 10-to-100-times slower than the photoreceptor microsaccades. We explain these eye-muscle-induced retina movements in the Introduction, confer their role in active vision in the Discussion, and add a new Fig. 7d to compare the slow eye-muscle induced drifts and oscillations to the ultrafast photoreceptor microsaccades.

(ii) Photoreceptor microsaccades are evoked by any wavelength the R1-R8 photoreceptors are sensitive to. Inside each *Drosophila* ommatidium, the *number of its light-activated photoreceptors and their combined contraction strength* set its photoreceptor microsaccade amplitude. In each ommatidium, R1-R8 photoreceptors are mechanically coupled and pivoted. Therefore, if only a single photoreceptor, say R1, is

light-activated, in the resulting microsaccade, the photomechanical R1 contraction alone is enough to move its neighbours (R2-R8) as well. As *Drosophila* R1-R6 and R7 cells are UV-sensitive, UV-light evokes larger photoreceptor microsaccades than, say, amber-light, which only activates R8y cells. These facts are now explained in the results and later discussed.

(iii) We now also show similar fused-rhabdom microsaccades from the honeybee apposition eye.

(iv) We now have added a new **Fig. 1(b)** and **Supplementary Video 4** to make the deep pseudopupil optical principles clear.

We are committed to providing a fair and constructive peer-review process. Do not hesitate to contact us if you wish to discuss the revision or if there are specific requests from the reviewers that you believe are technically impossible or unlikely to yield a meaningful outcome.

We expect major revisions of this nature to take around six months to complete, but appreciate that every situation is unique. Please take as long as necessary to address these concerns in full, including performing any additional experimental work required. We look forward to receiving your revised manuscript when it is ready and will not enforce any specific deadline. However, please bear in mind that if the revision process takes significantly longer than six months, we will need to confirm that nothing similar has been accepted for publication at *Communications Biology* or published elsewhere in the meantime.

Please do not hesitate to contact me if you have any questions or would like to discuss the required revisions further. Thank you for the opportunity to review your work.

Best regards,

Chao Zhou, PhD
Editorial Board Member
Communications Biology
orcid.org/0000-0002-8679-3413

Referee expertise:

Referee #1: neurogenetics, live brain imaging

Referee #2: Computational Neuroscience, active perception of visual system

Referee #3: insect neurobiology and behaviour

Reviewers' comments:

Reviewer #1 (Remarks to the Author):

BRIEF SUMMARY

1. This is a fascinating and comprehensive description of the approach and methods required to directly observe a new phenomenon in insect vision: photoreceptor microsaccades - the light-activated movement of the photoreceptor complex (rhabdomere) under a single lens of the compound eye. These photoreceptor microsaccades would be important to 'actively sampling' the visual scene in response to light increments. This, in turn, would enhance spatial resolution of the neural image formed by the brain. So now the insect compound eye is not a fixed sampling structure, but rather shows biomechanical responses across the retina that may be highly adaptive for spatial vision. The manuscript is fairly well written, although can be unnecessarily verbose and confusing in some sections. The approach is well detailed and documented. There are, however, numerous opportunities for improving the manuscript to enhance its readability and impact.

1. Our reply: Thank you. We have now carefully edited the manuscript to make its presentation simpler, clearer and better interconnected.

OVERALL IMPRESSION

2. My overall impression was that this is excellent: I want to build this system and observe these phenomena myself.

2. Our reply: Thank you 😊

SPECIFIC COMMENTS

CONCEPTUAL FRAMEWORK

3. The term "photoreceptor microsaccades" needs to be better distinguished from "microsaccades" as they are typically understood. Here, if I am reading this right, each ommatidium acts independently, responding to illumination locally at its lens, and producing a mechanical movement in the single direction specified morphologically by the axis of the rhabdomere. The global vector map of local PMS shows a 'flow field', but this field is not co-activated as a population. Rather, if you were to observe the full array of PMS in response to the movement of a natural scene, the PMS are activated by the spatially localised light levels. This is very different from retinal microsaccades, in which the entire retinal array is refreshed by the displacement of the full epithelium. For that matter, the term 'microsaccades' was meant to distinguish very small amplitude eye movements from much larger amplitude 'saccades' that reorient gaze. But here there are only 'photoreceptor saccades'...they are always 'micro'. Finally, saccades as they are traditionally referred to are unitary fixed action patterns. But here, the movements you characterise vary widely in amplitude. Are they really 'saccadic' in that sense?

3. Our reply: We thank the Reviewer for these useful comments. To make our results easier to understand, we now have added a new **Fig. 1** (attached below) and **Supplementary Video 4** to graphically show what the ultrafast photoreceptor microsaccades are, illustrating their (lateral and axial) micro-scanning movements inside each ommatidium. In the new **Fig. 1a** and *Introduction*, we now explain that the photoreceptor microsaccades are local ultrafast photomechanical photoreceptor contractions, that move R1-R8 photoreceptors - akin to a precisely oriented piston inside an ommatidial cylinder, as evoked by spatially-localised light intensity differences in the visual environment.

Fig. 1 | Active sampling by photomechanical photoreceptor microsaccades and the deep pseudopupil phenomena.
a, In the conventional static sampling theory, ommatidial facets set a compound eye's photoreceptor spacing, limiting the finest image details the eye could resolve. However, incoming light intensity changes in an ommatidium make its R1-R7/8 photoreceptors rapidly recoil axially and swing laterally. These so-called ultrafast photoreceptor microsaccades enable *Drosophila* to see the world in a finer resolution than its eyes' photoreceptor spacing, explained by the new active sampling theory. Left: *Drosophila* eye computer graphics (CG) model highlights the axial microsaccade component; R1-R7/8s first recoil and then slide towards the ommatidium lens. Right: concurrently, the light-activated R1-R7/8 also swing sideways (laterally). A local (incident) light intensity change evokes microsaccades only in those ommatidia facing the stimulus. If this happens in the frontal left and right eye ommatidia with overlapping receptive fields, their microsaccades are synchronous yet have mirror-symmetric lateral components. Meanwhile, elsewhere across the eyes, the photoreceptors stay still because the eye curvature and the ommatidial screening pigments block them from seeing the stimulus.
b, The optical principle of the deep pseudopupil (DPP). DPP is a virtual image of several distal R1-R7/8 rhabdomere tips (highlighted in blue, yellow and green for three nearby ommatidia), which align with the angle the eye is observed at while being ~10x-magnified by the ommatidial lens system. These virtual rhabdomere images are optically brought together when the microscopes' focal plane is set to ~200 μm under the eye surface (as shown in image 2). Because of the optical magnification, the rhabdomere tips, which appear deep inside the eye, are actually positioned at ~20 μm from the inner surface of the ommatidium lens.

About the term "photoreceptor microsaccade". This term was first coined in 2017 (Juusola et al., 2017), indicating ultra-fast unitary photomechanical jump-like sampling action of ommatidial photoreceptors, and has since gradually established itself in the literature. Although the photoreceptor microsaccade amplitudes vary, their dynamics are rather consistent for the given light intensity change. In fact, they vary

less than the spontaneous "retinal saccades" in immobilised insects (see new Fig. 7d, attached and our comments below).

Fig. 7 | DPP microsaccades in honeybee fused rhabdom apposition eyes.

a, Head-immobilised living honeybee prepared for the high-speed DPP imaging experiments.

b, Honeybee compound eye image series under antidromic infrared illumination, when the microscope gradually focusses inside the eye. First, the facet lenses appear (frames #3 and #4) before the DPP emerges as a small bright disk (frames #11 and #12).

c, High-speed imaging of photomechanical photoreceptor microsaccades. A characteristic slightly elongated DPP image of superimposed rhabdoms of neighbouring ommatidia. The DPP elongation is caused by the "ellipsoid" (non-spherical) honeybee eye shape. The scale bars give the actual rhabdom dimensions in ommatidia. The tested UV and green flashes evoked photomechanical photoreceptor microsaccades, seen as ultrafast lateral DPP "jump" and axial DPP darkening synchronised in time. The average DPP microsaccade to the green flash (green traces) was 1.54-times larger than to the UV flash (purple traces), consistent with the ommatidia having more 1.6-times more green photoreceptors (4) than UV (2.5) photoreceptors (inset).

d, Spontaneous slow eye-muscle-movement-induced (whole retina) DPP drifts were 10-to-100-times slower than light flash triggered photomechanical photoreceptor microsaccades. However, unlike in *Drosophila*⁵, the sluggish eye-muscle-activity shifted the Honeybee DPP positions over seconds and minutes more than the ultrafast (<200 ms) DPP microsaccade displacements superimposed on them.

About the photoreceptor microsaccade size. The photomechanical photoreceptor microsaccade amplitudes scale with each species' interommatidial angle (see new Fig. 8 attached below). And therefore,

photoreceptor microsaccades can also be large, even larger than typical spontaneous eye-muscle-induced whole retina movements (Kemppainen et al. *bioRxiv*, 2021) – as in wild-type *Drosophila* (Fig. 8a).

Fig. 8 | Fruit fly and Honeybee photoreceptor microsaccades scale with their receptive field half-width and interommatidial angle, presumably maximising retinal image acuity.

a, *Drosophila* photomechanical photoreceptor microsaccades typically shift their rhabdomeres 1-1.5 μm laterally (max $< 2 \mu\text{m}$), equating to $\sim 3\text{-}4.5^\circ$ receptive field movements in the visual space. R1-R6 photoreceptors' receptive field half-widths ($\Delta\rho$, retinal pixels) are between $4.5\text{-}6^\circ$, over-completely tiling up the visual space. The average interommatidial angle (ϕ) is $4.5\text{-}5^\circ$. Now consider this as image sampling by a digital camera. The spatial image information doubles when its sensor is moved, and two consecutive images are taken a 1/2-pixel apart and time-integrated for enhanced resolution. But if a pixel (receptive field) moves more, it eventually fuses with its neighbour (if that neighbour pixel was still – i.e. not detecting light changes). Because of this complete pixel fusion, acuity would decrease as the resulting neural image would contain fewer pixels. Therefore, by limiting micro-scanning to interommatidial angle, *Drosophila* can time-integrate a neural image, which greatly surpasses its compound eyes' optical limits.

b, Honeybee photoreceptor microsaccades shift their receptive field maximally $< 1^\circ$, approaching the eye's average interommatidial angle. Equally, such displacement is less than their average receptive field half-width (1.8°) in the front of the eye. As this active sampling strategy is broadly comparable to that of *Drosophila*, we predict that honeybee vision surpasses its compound eyes' static pixelation limit, similar to what we have shown for *Drosophila*.

As briefly mentioned above, in addition to photoreceptor microsaccades, intraocular muscles can shift the whole retina ("retinal saccades") within the left or right eyes' strict structural limits. However, these retina

movements in immobilised *Drosophila* and honeybee are 10-to-100-times slower than the photoreceptor microsaccades. We now consider in more detail the eye-muscle-induced retina movements in the Introduction, confer their role in active vision in the Discussion, and added a new **Fig. 7d** for that readers to compare these slow eye-muscle induced drifts and oscillations to the ultrafast honeybee photoreceptor microsaccades. Specifically, the combined effects of photoreceptor microsaccades and whole-retina movements in active sampling are now explained in a new *Discussion* section: "**Photoreceptor microsaccades and eye-muscle-induced whole retina movements mix**".

DETAILS (in approximate order of appearance in the text)

4. Microsaccade vector field movies: the blue is too low contrast against a black background. Similarly, the blue boxes on the microsaccade movie is very hard to see against the dark background, also in the static figures.

4. Our reply: Good point. We have now made these presentations stand out better.

5. In movies as well as figure panels: please add egocentric eye axes: frontal-backward, dorsal-ventral, otherwise it is very hard to orient. For example, in your movie, what does "Dorsal view" mean? Are we viewing the vector field from the dorsal perspective, in which case frontal (anterior) is upward? For Anterior view, is dorsal upward? Label these everywhere.

5. Our reply: We have now added this information in the figures and videos as needed.

6. Make clear in the text that you are stimulating the WHOLE EYE with UV light. In which case, what is the difference between "global" and "local" experiments?

6. Our reply: We have now moved the respective text sections around to eliminate this confusion from arising. We did not stimulate the whole eye in these experiments but presented a *local* light field stimulus to local photoreceptors. Because of the eyes' spherical shape with each ommatidium being isolated by pigment cells, the photoreceptors that participated in the DPP microsaccades see only local light changes. We have made a new graphical video animation to make this point easier to understand (new **Supplementary Video 4**). The only "easy" *global* way to activate all the photoreceptors *homogeneously* at once is to use high-intensity X-rays (as we have shown previously using DESY and ESRF synchrotrons; Kempainen et al. *bioRxiv*, 2021).

7. WOW! Note the highly prominent antennal movements in your photoreceptor microsaccade movie! Cite paper from Dickinson lab (J. Neuroscience?) showing optomotor antennal movements - are these phenomena related? Discuss!

7. Our reply: We have recently published separate analyses about antennae movements (see Kempainen et al. *bioRxiv*, 2021; see its supplementary figure S35). Those recordings clarified that while the light flashes triggered antennae casting in some flies, this always happens after the photomechanical photoreceptor microsaccades, at least 40-50 ms later. The same publication further shows that separately puffing air to the antennae made no difference in the observed photoreceptor microsaccade dynamics (see its supplementary figure S36). Following the Reviewer's suggestion, in the current manuscript, we now briefly consider how additional antennae casting could benefit the integration of multisensory information in *Discussion* (also citing the suggested JNS paper). In addition, we have added a new **Supplementary Video 5** that shows high-speed recordings from five wild-type and five spam *Drosophila*. The video makes it easy to see that sometimes some flies generate antennae casting after light stimulation, but at other times they do not. So antennae casting is at least partly voluntary.

8. Do other wavelengths of light activate photoreceptor microsaccades?

8. Our reply: We have now added in the results: "A flash of any wavelength R1-R8 photoreceptors are sensitive to (~300 to ~650 nm)^{22, 26} evokes a photoreceptor microsaccade⁵. Inside an ommatidium, the number of light-activated photoreceptors and their combined contraction strength set its photoreceptor microsaccade amplitude⁵. Because these photoreceptors are mechanically coupled and likely pivoted⁵, it only takes one (say R1) to be light-activated, and its contraction alone can move its neighbours (R2-R8) too⁵ (**Supplementary Video 4**). As *Drosophila* R1-R6 possesses the sensitising UV-pigment and R7s are UV-

sensitive^{22, 26}, a UV-flash evokes larger photoreceptor microsaccades than, say, an amber-flash, which only activates R8y cells⁵. Therefore, in *Drosophila* experiments, UV-flash is a good choice of stimulus."

9. L42: ...shift one eye's sampling matrix globally relative to...

9. Our reply: now corrected.

10. Introduction: you refer to the photoreceptor saccades as being a local phenomenon, whereas intraocular muscles act globally. Do you know for sure that each rhabdomere acts independently? The field of illumination is large (this point also needs to be clarified within the narrative), so although you only image individual rhabdomeres via DPP, others may be co-activated, right? Or are you speculating that the spatial distribution of photoreceptor microsaccades mimics the local light stimuli generated by the spatial arrangement of light-dark edges moving across the eye?

10. Our reply: See our response to point 8 above. Please see the new **Supplement Video 4**, which we specifically produced to answer these questions.

11. Figure 1: (a,b) indicate (estimate) the imaging plane on the vertical ommatidia drawings.

11. Our reply: the old **Fig. 1** is now a new **Fig. 3** (attached below). This imaging plane is the same as shown in the new **Fig. 1b**. The same information is now mentioned in **Fig. 3** legends.

12. Figure 1: You need to show raw data, raw images. It is fine to also show your CG modeled images alongside, but not in place of. Similarly, L63: "virtual images" are inappropriate for a methods paper, at least as the narrative stands in its current form. Most readers will become cross with this approach.

12. Our reply: We have produced new **Figs 1** and **3**. These illustrate the DPP principle and link the simulation to the raw imaging data shown in **Fig. 4** and **Supplementary Videos 2, 4** and **5**.

13. Figure 2: Is there a particular reason that the stereoscope is mounted horizontally?

13. Our reply: We have now added in the methods: "Note that the stereo microscope's horizontal mounting is not critical but resulted from our earlier design choices. By having an upright rotation stage, this configuration works well for the binocular compound eye measurements."

14. Why do you choose UV wavelength to activate photoreceptor microsaccades (PMS)? Do other wavelengths activate the PMS? I realise that you have another paper under review, posted to bioRxiv, which I read through in order to better understand this one. But this paper has to 'stand alone' and so there needs to be a bit more conceptual unpacking, I think. Also, does full-field illumination activate ALL of the PMS together? Does screening pigment localise them, particularly for shorter wavelength light? Maybe some of these issues are best treated in the results-focused companion to this method-focused one.

14. Our reply: Please see our response to point 8.

15. Figure 3: please re-emphasise that the DPP that you image is a SINGLE rhabdomere from a SINGLE ommatidium, but that the imaging optics magnify it to occupy many more than one ommatidium in the image. Do I have this right? Please make this clear in the text.

15. Our reply: Theoretically (and also almost certainly in reality), for a stereomicroscope with NA = 0.11, the observed DPP is a summed up image of seven rhabdomeres, with 50% summation from up to 17 ommatidia (please see **Supplementary Table 1** and the new **Supplementary Video 4** that shows this graphically).

16. Figure 3a-e: blue rectangle is hard to see against dark image

16. Our reply: Now corrected.

17. Figure 3c: perhaps show ONE spam DPP image?

17. Our reply: Done.

18. Figure 3d caption: missing word "...appear as..."

18. Our reply: corrected as suggested.

19. Figure 4: the supplement video showing photoreceptor microsaccades is absolutely convincing. Yet, some readers will not wish to look at the supplement (I know....) so I would suggest a temporal sequence of single frames showing a photoreceptor microsaccade.

19. Our reply: As suggested, this is now added in new **Fig. 5c**.

20. Figure 4c,d: indicate D-V, A-P axes

21. Our reply: Now rectified.

21. L126: This point, that the axis of the photoreceptor microsaccade aligns with the rhabdomere axis is crucially important as it speaks to mechanism. You should show a superposition of Fig 3c and 4c to show this.

21. Our reply: As requested, please see a new **Supplementary Video 3** that shows this relationship dynamically.

...

22. SPAM mutant: why don't you simply test an insect with an apposition eye instead of the mutant *Drosophila* with a fused rhabdom? Do you expect that a fused rhabdom eye would have more or less disruption of the photoreceptor microsaccadic orientation map?

22. Our reply: Please see the new **Fig. 7** and results section: "Photoreceptor microsaccades in honeybee fused rhabdom apposition eyes".

23. L133 extra word "the"

23. Our reply: "the" now omitted.

24. L136: the use of 'global' vs 'local' is very confusing here. You could clear this up in the narrative. "In order to obtain more robust results, we re-sampled photoreceptor microsaccades 25 times from a single ommatidium, located at ____." Also at L151 "We characterised variation across individual flies by re-sampling from a single retinal location..." This issue comes up again on L180; in fact ALL of your measurements are "local". You only image ONE DPP at a time.

24. Our reply: these sentences are now deleted.

25. L182-183: I do not follow the premise on the lever action here. Can you elaborate a bit?

25. Our reply: We now write: "...The fused rhabdom rhabdomeres embrace each other and rigidify. Therefore, during microsaccades, they would have less flexibility to move sideways than the open wild-type *Drosophila* rhabdomeres."

26. Here, this is a methodology paper, in which you do not present evidence for "active sampling". In fact, one could conclude from the data presented here that the DPP microsaccades are reflexive, particularly since they are always accompanied by antennal movements in the video. I wonder whether it makes sense to revise this to de-emphasise the interpretation of active sampling, and that the DPP movements are 'microsaccades' at all. You merely present evidence for light activated DPP movements.

26. Our reply: We now write in *Discussion*: "Photomechanical photoreceptor microsaccades are not reflex-like uniform contractions^{4,5}. Instead, at each moment, they actively and continuously auto-regulate photon sampling dynamics by moving and narrowing the photoreceptors' receptive fields in respect to environmental light contrast changes to maximise information capture⁴. These dynamics rapidly adapt to immediate light history and are different at dim and bright conditions and to positive (light increments) and negative contrasts (light decrements)^{4,5}. So, from the viewpoint of sampling theory, photoreceptor microsaccades are not passive but constitute a form of ultrafast *morphodynamic* active sampling^{4,5}. However, in this study, we only examined photoreceptor microsaccades in one stimulus condition; to bright light flashes after prolonged dark adaptation."

Moreover, antennae casting is under voluntary control: some flies show this sometimes, others never; see our response to point 7 above. Finally, please see the new discussion section for further insight: Photoreceptor microsaccades and eye-muscle-induced whole retina movements mix.

27. L199 extra word "how" 27. Our reply: "how" is now omitted.

Reviewer #2 (Remarks to the Author):

1. This manuscript describes a novel technique to measure photomechanical microsaccades in compound eyes. The authors utilise high-speed imaging to characterise the contraction of the photoreceptors to flashes of light in both open and fused rhabdom eyes. This approach enables the authors to investigate how these contractions differ between the two eye types and highlight an additional strategy by which compound eyes actively sample a visual scene in a way that resonates with the natural visual flow.

This study is interesting and introduces a technique that enables further investigations in active vision not only through self-motion but through a mechanism that is intrinsic to the compound eye. The perceptual benefits of such contractions have been reported previously by this group (Juusola et al, 2017, eLife); and the results here provide a way to investigate in much greater resolution the characteristics of both the global and local motion patterns. The methods are thoroughly described, and software approaches are openly available. Overall, the results are sound, and I look forward to the future exciting results that such an approach can provide to the field of active vision.

1. Our reply: Thank you 😊

Specific Comments:

2. Line 80: "bellow" should be "below"

2. Our reply: Corrected as suggested.

3. Figure 4 Caption: "d, Mean microsaccade vector map of spam -flies-"

3. Our reply: Corrected.

4. Figure 4e-f and lines 119-123: It is unclear what "absolute difference" as collinear and perpendicular is quantifying, and how it is different from the directional difference. Please elaborate or include a definition in _____ methods.

4. Our reply: Corrected to angular difference.

5. Line 199: "Both of these components shape the way -x-how-x- a photoreceptor encodes ..."

5. Our reply: "how" removed.

6. Figure 5: "f-j" should be "f-i"

6. Our reply: Now corrected.

7. The speculation and Discussion following the analysis of variability of repeated responses and variance of displacement (Fig. 5j-k) are difficult to follow. How does such variation indicate purposeful regulation of these contractions?

7. Our reply: We now mention one concrete example, citing our previous publications: "Such variability could result, for instance, from the slow eye-muscle-induced whole retina drifting⁵, structurally fluctuating the local rhabdomere tension (their anchoring and pivoting dynamics) from one trial to another⁵. We further note the real benefit of the local photoreceptor microsaccades' stochastic variability, irrespective of the cause, is that it effectively removes aliasing from the retinal sampling matrix^{4,5,30}."

Reviewer #3 (Remarks to the Author):

This paper describes a technique to measure lateral photoreceptor movements during light stimulation in vivo for *Drosophila*, and potentially other compound eyes. The authors combine deep pseudopupil imaging (with infrared, non-stimulating light), high-speed cameras, and a goniometer to quantify the photoreceptor microsaccades that result from membrane contraction during light absorption. They compare rhabdomere movements in wild-type *Drosophila melanogaster*, with movements in a mutant possessing a fused rhabdom, similar to typical apposition eyes.

Major:

1. A big conclusion of this paper is that "photoreceptor microsaccades are not limited to the open rhabdom eye design but can also occur in fused rhabdom eyes", such as honey bees, and the vast array of apposition eyes that aren't neural superposition. However, a mutant fruit fly certainly doesn't demonstrate this convincingly, as it may display this ancestral trait (a fused rhabdom) in a very different way from species that develop this way naturally (without a mutation). This is like claiming the presence of eyes on a blind mutant shows that eyes occur in other animals that don't have vision. The model system doesn't have quite this much power---even if it is likely to be true, I need to see it in at least one animal with proper fused rhabdoms to believe this claim.

1. Our reply: A fair point. We agree and therefore have now confirmed this result also in the honeybee fused rhabdom apposition eyes. See new Fig.7.

2. A figure showing the light paths for deep pseudopupil imaging would be quite helpful for most readers trying to follow the technique. Figure 1 shows DPP images nicely, but it could easily also incorporate a ray trace illustration, which would help readers interpret, for example, the central left and right images in Figure 3c, which are otherwise confusing. Figure S2 has something like this, or something like 7.11 in *Animal Eyes*. It would help readers understand, for example why "to observe a well-defined, clear DPP pattern in dipteran eyes requires precisely 367 organised rhabdomeres across the neighbouring ommatidia (In 66)"

2. Our reply: We agree and therefore have included a new Fig. 1b that shows using beam-tracing how the microscope system collects the virtual deep pseudopupil image. In addition, we have produced a new Supplementary Video 3 to help the readers better visualise these dynamics and the underlying physics concerning the used microscope system.

3. The paper also lacks a good image showing movement of the rhabdomeres. They are characterized in figure 4, but I was surprised by the lack of a clear figure showing the rhabdomeres before and during the UV light flash, to illustrate where the vectors in 4c and 4d actually come from.

3. Our reply: We have added a new Fig, 5c to illustrate this.

4. An easy issue to address is that the many figure panels are far too small, with axis labels and arrows much smaller than the body text, and severely pixelated when I zoom in to see better. In 3b, 3e, 4b, and 5b I can barely make out the DPP images, which ought to be the highlight of this paper.

4. Our reply: We have made the figures bigger and now consistently use larger font sizes.

Minor:

5. In 25 "These results show different ways how vision converts space into time" -I understand what you are getting at, that spatial light patterns in the environment become temporal electrical signals of a photoreceptor, but "converts space into time" is too compact, and makes it sound like a physics paper on relativity.

5. Our reply: We now write: "These results show different ways compound eyes initiate the conversion of spatial light patterns in the environment into temporal neural signals, and highlight how this active sampling can evolve with insects' visual needs."

6. In 48 "atomic force microscopy (AMF)"

-AMF or AFM?

6. Our reply: Corrected to AFM.

7. In 107 figure 4.-4b on the right appears to show the microsaccade movement, but this isn't described in the legend, only the rotation relating to the right image. Are the top and bottom right images wt and spam?

7. Our reply: This figure is now Fig. 6. We now explain this in the legends.

8. -4e and 4f compare the absolute and directional difference of vector maps, 4c and 4d. I would have thought this would be a point by point difference in vector length for 4e and difference in angle for 4f. But the scale bar in 4e is labeled 'perpendicular' to 'colinear', which sounds like a directional difference to me. I need clarification here.

8. Our reply: These plots are now renamed to show the angular difference.

9. In 125 "These results demonstrate active sampling - by photomechanical photoreceptor microsaccades - happening in fused rhabdom eyes." -How do the results demonstrate active sampling? They demonstrate motion in response to light, but this may, in principle, be non-functional in the spam mutant, just a vestigial response that does not affect, or maybe even reduces sampling in the fused rhabdom.

9. Our reply: The photoreceptor microsaccades are *photomechanical*, self-regulating the incoming light information sampling (see Juusola et al., 2017). Therefore, *spam* photoreceptor microsaccades indicate functional phototransduction (PIP₂ cleavage), demonstrating active sampling. Accordingly, during microsaccades, *spam* photoreceptors generate normal voltage responses and histaminergic synaptic transmission to the brain. We have shown this in our recent other publication in Kempainen et al., 2021 (see supplementary figure S6 there).

10. In 134 figure 5 -'Displacement' is usually a term for a vector, but except for the images in 5c which show an arrow, the authors are using it as a scalar (5f, 5k, legend), the magnitude of displacement. Then in the main text they use the term 'displacement amplitudes', seeming to indicate a vector again. This should be made consistent, and if the authors decide not to use 'displacement' as a vector, explain early on in the text.

10. Our reply: We agree and have changed this to microsaccade amplitude.

REVIEWERS' COMMENTS:

Reviewer #1 (Remarks to the Author):

The authors have done a very nice job addressing the referee comments. I have no further criticisms.

Reviewer #3 (Remarks to the Author):

The authors have addressed all my concerns, and have added substantial content and clarification to satisfy the reviewers. I'm impressed with reworked figures, the new figure 1, and especially important is the addition of honeybee data. I can certainly recommend accepting the manuscript.